# Variability in runoff fluxes of dissolved and particulate carbon and nitrogen from two watersheds of different tree species during intense storm events

Mi-Hee Lee[1], Jean-Lionel Payeur-Poirier[2], Ji-Hyung Park[3], Egbert Matzner[1]

[1]Department of Soil Ecology, University of Bayreuth (BayCEER), Dr.-Hans-Frisch-Straße 1-3, Bayreuth, D 95448, Germany
[2]Department of Hydrology, University of Bayreuth (BayCEER), Universitätsstrasse 30, Bayreuth, D 95447, Germany
[3]Department of Environmental Science and Engineering, Ewha Womans University, Ewhayeodae-gil 52, Seodaemun-gu, Seoul, 03760, South Korea

*Correspondence to*: Egbert Matzner (egbert.matzner@uni-bayreuth.de)

**Abstract.** Heavy storm events may increase the amount of organic matter in runoff from forested watersheds as well as the relation of dissolved to particulate organic matter. This study evaluated the effects of monsoon storm events on the runoff fluxes and on the composition of dissolved (< 0.45 μm) and particulate (0.7 μm to 1 mm) organic carbon and nitrogen (DOC, DON, POC, PON) in a mixed coniferous/deciduous (mixed watershed) and a deciduous forested watershed (deciduous watershed) in South Korea. During storm events, DOC concentrations in runoff increased with discharge, while DON concentrations remained almost constant. DOC, DON and $NO_3$-N fluxes in runoff increased linearly with discharge pointing to changing flow paths from deeper to upper soil layers at high discharge, whereas nonlinear responses of POC and PON fluxes were observed likely due to the origin of particulate matter from the erosion of mineral soil along the stream benches. The integrated C and N fluxes in runoff over the 2 months study period were in the order; DOC > POC and $NO_3$-N > DON > PON. The integrated DOC fluxes in runoff during the study period were much larger at the deciduous watershed (16 kg C ha$^{-1}$) than at the mixed watershed (7 kg C ha$^{-1}$), while the integrated $NO_3$-N fluxes were higher at the mixed watershed (5.2 kg N ha$^{-1}$) than at the deciduous watershed (2.9 kg N ha$^{-1}$). The latter suggests a larger N uptake by deciduous trees. Integrated fluxes of POC and PON were similar at both watersheds. The composition of organic matter in soils and runoff indicate that the contribution of near surface flow to runoff was larger at the deciduous than at the mixed watershed. Our results demonstrate different responses of particulate and dissolved C and N in runoff to storm events as a combined effect of tree species composition and watershed specific flow paths.

**Keywords.** Dissolved organic carbon, Dissolved organic nitrogen, Particulate organic carbon, Particulate organic nitrogen, Monsoon storm, Forested watershed

## 1 Introduction

As much of the dissolved organic matter (DOM) in aquatic systems originates from soil derived organic matter, the export of terrestrial carbon (C) and nitrogen (N) into aquatic environments is a primary link between these systems (Bauer and Bianchi,

2011; Bianchi, 2011; Camino-Serrano et al., 2014; Canham et al., 2012). The export of terrestrial C and N occurs in the form of dissolved and particulate organic carbon and nitrogen (DOC, DON, POC, PON). Particulate organic matter can be operationally classified into fine (0.1 to 63 μm) and coarse (63 μm to 2 mm) fractions (Richey, 2005). The export of POC was in some cases the major C export in stream (Dhillon and Inamdar, 2013; Jung et al., 2012; Kim et al., 2010; Lloret et al., 2013). On the contrary, DOC was reported as the dominant organic C form in a temperate headwater catchment (Johnson et al., 2006), a tropical rainforest catchment (Bass et al., 2011) and for several large tropical watersheds such as Amazon, Orinoco, Parana and Mengong (Lloret et al., 2013).

In regions with seasonally large differences in precipitation, most of the annual organic C export from forested watershed to steams is driven by heavy storm events with cyclones and hurricane (Dhillon and Inamdar, 2013; Lloret et al., 2013). Such conditions are pronounced in the Korean peninsula in which the monsoon season (Jeong et al., 2012; Kim et al., 2010), represented 52% and 83% of the annual DOC and POC runoff fluxes. During storm events, a change in hydrological flow paths in watersheds has been often observed from deeper to upper soil layers (Bass et al., 2011; Sanderman et al., 2009; Singh et al., 2014). Surface flow-inducing storm events can alter the fluxes and concentrations of DOC and POC in runoff by shifting preferential flows through macropores, surface runoff and lateral flow (Katsuyama and Ohte, 2002; Kim et al., 2010; McGlynn and McDonnell, 2003).

In case of organic N export, DON was the major form of N in runoff from pristine forested watersheds (Alvarez-Cobelas et al., 2008; Frank et al., 2000; Kaushal and Lewis, 2003; Pellerin et al., 2006; Yates and Johnes, 2013). Only few data are available on the partitioning of DON and PON fluxes in runoff from forested watersheds, like Inamdar et al. (2015). They reported that particulate N composed 39-87% of the storm event N export. The question remains open, if organic N in runoff – either dissolved or particulate – from forested watersheds behaves similar to organic C or not. Some studies reported that concentrations of DON and DOC correlated strongly (von Schiller et al., 2015), but also weak relationships were found (Singh et al., 2015).

Considering an effect of watershed characteristic, tree species might influence the export of DOM from forested watersheds. DOM from coniferous litter generally comprises more refractory (e.g. hydrophobic acid, lignin) and aromatic compounds, and a relatively larger proportion of high molecular weight compounds than DOM from deciduous litter. It is also more acidic than DOM from deciduous litter (Don and Kalbitz, 2005; Hansson et al., 2011; Kiikkilä et al., 2013). Moreover, higher DOC and DON concentrations were found in oak, beech, and silver birch forest floors compared to Norway spruce, Douglas-fir, and Scots pine (Smolander and Kitunen, 2011; Trum et al., 2011). Amiotte-Suchet et al. (2007) found higher annual DOC concentrations and fluxes in runoff at a deciduous forested watershed than at a watershed dominated by coniferous species.

As a result of global warming, heavy storm events have occurred more frequently and became stronger in recent decades (IPCC, 2013). Furthermore, forest management, namely the selection of tree species, might influence the export of organic matter from forested watersheds. Understanding the influence of both drivers is needed for a better prediction of the link between terrestrial and aquatic ecosystems and to support an efficient downstream water quality management. The goal of

this study was thus to investigate the influence of tree species and heavy storm events on the fluxes of dissolved and particulate forms of C and N from a mixed coniferous/deciduous and a deciduous forested watershed in South Korea during the 2013 monsoon season.

## 2 Materials and methods

5 **2.1 Study area and site**

The Lake Soyang basin area (Figure 1) is located in the upstream region of the Han River, which is the main source of drinking water for about 23 million citizens of South Korea (Lee et al., 2013; Park et al., 2010). The average annual temperature of the Lake Soyang watershed in western Gangwon-province is 11ºC with monthly average temperature ranging from -5ºC in January to 24ºC in August (Korean meteorological administration). Annual precipitation ranges from 1200 to 10 1500 mm and the summer monsoon usually accounts for 50 to 60% of the annual precipitation (Park et al., 2010; Seo et al., 2011). Korean mountainous forests are mostly composed of deciduous forests representing 47% of the total forested area (38% coniferous forest, 12% mixed coniferous and deciduous forest) and most of the broadleaved forests of South Korea are distributed within the Gangwon province (Korea forest research institute, 2013).

The mixed coniferous/deciduous forested watershed (mixed watershed; Figure 1) is located in Seohwa, the Gangwon 15 province (38°12′N, 128°11′E, 368 to 682 m above sea level). The area of the mixed watershed (Table 1) is 15.6 ha with 6.1 ha of coniferous forest (39%) and 9.5 ha of deciduous forest (61%). Two research plots as one in the coniferous part (MC plot) and the other one in the deciduous part (MD plot) were established. The slope of the mixed watershed as obtained from a digital elevation model ranges from 4.0 to 41° with an average of 28°. The lower part of the mixed watershed is dominated by coniferous species, including *Larix kaempferi* (Lamb.) Carr. (Japanese larch) and *Pinus densiflora* Siebold & Zucc. 20 (Japanese red pine). The upper part of the mixed watershed is dominated by deciduous species, such as *Juglans mandshurica* Maxim. (Manchurian walnut), *Acer pictum* subsp. *mono* (Maxim.) H.Ohashi (Mono maple), *Quercus dentata* Thunb. (Daimyo oak), *Tilia amurensis* Kom. (Lime tree) and *Ulmus davidiana* var. japonica (Rehder) Nakai (Japanese elm).

The slope direction of the coniferous part at the mixed watershed is towards the MD plot. Lateral flow from the coniferous part to the MD plot can only influence deeper soil solution characteristics as near surface flow was never observed. Our data 25 (see results) indicate significant quality differences of soil solutions between the MD and MC plots which suggest only a minor influence on soil solution chemistry at the MD plot from lateral flows. Furthermore, the quality parameters of soil solutions at the MD plot were similar to those of the DD plot, the latter being not influenced by lateral flows from coniferous sites. Thus, it is unlikely that the MC plot did affect the MD plot.

30 The deciduous forested watershed (deciduous watershed; Figure 1) is located in Haean, the Gangwon province (38°15′N, 128°7′E, 586 to 1005 m above sea level). The area of the deciduous watershed (Table 1) is 39 ha and is covered by various deciduous species. A research plot as deciduous plot (DD plot) was established in this watershed. The slope of the deciduous

watershed ranges from 4 to 53° with an average of 24°. The dominant tree species are *Juglans mandshurica* Maxim. (Manchurian walnut), *Acer pictum* subsp. *mono* (Maxim.) Ohashi (Mono maple), *Quercus dentata* (Daimyo oak), *Quercus mongolica* (Mongolian oak) and *Fraxinus rhynchophylla* (Korean/Chinese ash). The average age of trees in the two watersheds is about 35 years. The distance between the two watersheds is ca. 6 km.

## 2.2 Experimental design

### 2.2.1 Water sampling

Bulk precipitation samplers (n=2) were installed at each watershed in an open area located ~100 m from the plots. Throughfall collectors (n=5) under the canopy were equipped with filters to prevent large particles from entering. Forest floor leachate was collected beneath the organic layer along the slope side using zero tension lysimeters (n=5) of 185 $cm^2$ made of acrylic material. Soil solution was collected at a depth of ~50 cm with suction lysimeters (n=5) made of ceramic cups.

Before storm events in June 2013, throughfall, forest floor leachate and soil solution were collected at about weekly intervals, and runoff samples were collected 2-3 times per week. During storm events in July 2013, throughfall, forest floor leachate and soil solution was collected after each storm event so that these samples represent cumulative water samples during the entire storm event. In case of runoff, samples were taken in July 2013 at the weir using automatic collectors (6712 Portable Sampler, Teledyne Isco Inc., Lincoln, NE, USA) before, during, and after each rain event at intervals of 1 or 2 h. Discharge at the outlet of the watersheds was measured by a v-notch weir. During routine runoff sampling, water temperature, pH and electrical conductivity was measured in situ. Water samples were cooled at 4°C and then were filtered (see 2.4) within 2 days after sampling. Filtered solution samples were frozen for 1 month until further analysis of water quality and quantity.

Precipitation data (total and hour unit; Table 2) at the study area were used from the automatic weather station of the Korean meteorological administration at the point 'Seohwa 594' and 'Haean 518' for the mixed watershed and for the deciduous watershed, respectively. Those data were also comparable to ours from bulk precipitation measurements at the field sites.

### 2.2.2 Soil sampling

The total stock of organic horizons (Oi: slightly decomposed recognizable litter, Oe: moderately decomposed fragmented litter, Oa: highly decomposed humic material) was collected at each plot in a 20 × 20 cm frame with 10 replicates. The average thickness of Oi and Oe+Oa was 1.2 and 1.5 cm at the MC plot, 2.5 and 3 cm at the MD plot, and 2.3 and 2 cm at the DD plot, respectively. Mineral soil samples were collected from 3 pits at each plot in 10 cm depth layers down to 50 cm depth. In case of the DD plot, the sampling of mineral soil was not possible deeper than 40 cm depth due to massive rock. Before the analyses, soil samples were air-dried and crushed to pass through a 2 mm sieve. Soil pH was measured from a solution of a soil to solution (0.01 M $CaCl_2$) ratio of 1:2.5 after shaking for 2 hours. Total C and N contents were analyzed using an elemental analyzer (vario MAX CN, Elemental, Germany). Soil texture was determined by sedimentation.

**2.3 Calculation**

**2.3.1 Fluxes of C and N in runoff**

In June 2013, before the monsoon storm events, the fluxes of DOC were calculated on a weekly basis by multiplying the DOC weekly mean concentration in runoff by the weekly mean discharge. The concentrations of DON, $NO_3$-N, POC and PON in runoff were partly below the detection limit. Concentrations less than detection limit were observed in 5-8% of the measurements in runoff during the July events. The detection limits were applied to the calculation of export fluxes as 0.03 mg DON $L^{-1}$, 0.5 mg $NO_3$-N $L^{-1}$, 0.003 mg POC $L^{-1}$, 0.0003 mg PON $L^{-1}$. During the period of storm events in July 2013, the fluxes of DOC, DON, $NO_3$-N, POC and PON in runoff were computed at 1 or 2 h intervals by multiplying the measured concentrations with the corresponding discharge. During the monsoon season the rainfall was not continuous on all days but with intermittent gaps. The most lasting rainfall events were identified as storm events with more than a day interval between each storm event.

**2.3.2 Statistic for origin of DOM and POM**

The normality of data was tested with the Shapiro-Wilk Test. When the normality was assured, the Holm-Sidak Test was used for both pairwise comparisons and comparisons versus a control group. When the normality test failed, the Dunn's Test was used for all pairwise comparisons and comparisons against a control group with rank-based-ANOVA.

**2.4 Chemical analyses**

After filtration through a pre-rinsed cellulose acetate membrane filter (0.45 µm, Whatman), the concentrations of DOC and total dissolved nitrogen (TDN) in water samples were measured by a total organic carbon analyzer (TOC-CPH, Shimadzu, Japan). DON concentration was calculated as the difference between total nitrogen and mineral-N ($NO_3^-$ + $NH_4^+$). Nitrate and ammonium concentrations were measured by flow injection (FIA-LAB; MLE, Dresden, Germany). Nitrite was not measured because concentrations were negligible in soil solutions and runoff.

In this study, the POC and PON fraction is defined as the size class 0.7 µm to 1 mm. Samples were filtered through a 1 mm mesh to remove larger particulate materials and then finally filtered through a pre-rinsed 0.7 µm pore size glass filter (GF/F, Whatman). Before using the glass filters, the filters were pre-combusted at 450°C to remove any organic material. The residues of particulate material on the GF/F filters were analysed for POC and PON after drying at 65°C using an elemental analyser (Carlo Erba1108, Milano, Italy) coupled to a ConFlo III interface and an isotope ratio mass spectrometer (Finnigan MAT, Bremen, Germany). DOC and POC cutoff limits as 0.45 and 0.7 µm were unmatched in this study because of practical reasons and the unmatched fraction is considered negligible.

The absorption spectra of DOM were obtained at wavelengths from 200 to 600 nm using a UV-visible spectrophotometer (DR5000, HACH). Specific ultraviolet absorbance ($SUVA_{280}$) values were determined by UV absorbance at 280 nm divided by the DOC concentrations and multiplied by 100.

For fluorescence excitation-emission matrices, fluorescence intensities were recorded with a luminescence spectrometer (LS-55, Perkin-Elmer, USA) following the method of Baker (2001), Chen et al. (2007), and Hur and Cho (2012). Excitation and emission slits were both adjusted to 10 nm. DOM samples were diluted under the ultraviolet absorbance of 0.1 at 280 nm to avoid inner-filter correction, and then were adjusted to pH 3.0 for the fluorescence measurements. The fluorescence intensities of all samples were normalized to units of quinine sulfate equivalents. The humification index (HIXem) was calculated by dividing the emission intensity from 435 to 480 nm region by intensity from 300 to 345 nm (Zsolnay et al., 1999). Fluorescence characteristics of water samples were interpreted as fulvic-like fluorescence (FLF), humic-like fluorescence (HLF) and protein-like fluorescence (PLF) (Fellman et al., 2010; Singh et al., 2014).

After filtration (0.45 μm, Whatman), water samples were freeze-dried to measure $^{13}$C and $^{15}$N isotope abundances of DOC and TDN using an elemental analyzer (Carlo Erba1108, Milano, Italy) coupled to a ConFlo III interface and an isotope ratio mass spectrometer (Finnigan MAT, Bremen, Germany).

## 3 Results

### 3.1 Soil and hydrological characteristics

The morphology of the organic layers at the MC, MD, and DD plots were similar, representing a moder-like organic layer, with distinct Oi-layers and less distinct Oe and Oa-layers. However, the depth of O-layer in the MC plot (ca. 3 cm) was thinner than in the MD and DD plot (ca. 4-5 cm). The typical soil type at both watersheds is Dystric Cambisols (FAO, 2014). Soil texture at all plots ranged from 40-44%, 30-38% and 18-22% for sand, silt and clay, respectively. The C content of the organic layers at all plots ranged from 45 to 48% in the Oi and from 34 to 38% in the Oe+Oa layers. The C/N ratio at all plots decreased from the organic layer (20-29) to the mineral soil (10-12) down to 40-50 cm depth. The soil $\delta^{13}$C and soil $\delta^{15}$N values significantly increased with soil depth from -29 to -24‰ and from 0 to 8‰, respectively (Figure 2).

The average discharge in June 2013 before storm events was 0.03 mm h$^{-1}$ at the mixed watershed and 0.06 mm h$^{-1}$ at the deciduous watershed (data not shown). The total amount of precipitation in July was slightly higher at the deciduous watershed (367 mm) than at the mixed watershed (313 mm; Table 2). Also, the intensity of precipitation in July was larger at the deciduous watershed than at the mixed watershed. Similar to precipitation data, the mixed watershed had less maximum discharge and also slightly lower discharge before start of a storm event than the deciduous watershed.

### 3.2 Concentrations of carbon and nitrogen in runoff during storm events

The increase of the DOC concentrations in runoff with discharge was steeper at the deciduous watershed (e.g. 1.9 to 6.9 mg C L$^{-1}$ on July 8$^{th}$, 2013) than at the mixed watershed (e.g. 1.0 to 3.7 mg C L$^{-1}$ on July 8$^{th}$, 2013) (Figure 3a). In contrast, the DON concentrations in runoff from both watersheds were independent of discharge (Figure 3b). The highest concentration of DOC and DON in runoff was observed during the earlier storm events (Table 2). The NH$_4$-N concentrations were at any time negligible (< 0.05 mg N L$^{-1}$).

At discharges from ~1 to 9 mm h$^{-1}$, higher concentrations of POC and PON in runoff were found (Figure 3d,e). For example, the POC concentration in runoff from the mixed watershed was as high as 10.7 mg C L$^{-1}$ at the largest discharge of 9 mm h$^{-1}$. At the deciduous watershed, the POC concentration in runoff reached a maximum of 8.6 mg C L$^{-1}$ already at 3 mm h$^{-1}$ discharge during the first storm event (Figure 3d, Table 2). The following more intense storms did result in lower POC concentrations. The pattern of POC concentration coincided with those of PON (r=0.99).

The runoff DOC concentrations in response to discharge had a clockwise hysteretic loop with higher concentrations on the rising than on the falling limb (Figure 3a). No hysteretic loops were observed for DON, POC and PON (Figure 3b,d,e).

The DOC/DON ratio in runoff ranged from 5 to 60 (Figure 3c). The DON concentrations lower than 0.05 mg N L$^{-1}$ were not considered for calculation of the DOC/DON ratios. In response to increased discharge, the DOC/DON ratios were stable at the mixed watershed, while there was a tendency for increasing in the DOC/DON ratios with discharge at the deciduous watershed. On the contrary, there was no response of the POC/PON ratio to discharge. Unlike to the DOC/DON ratio, the POC/PON ratio ranged narrowly from 10 to 20 at both watersheds (Figure 3f) with an average of 12 at the mixed and 13 at the deciduous watershed.

## 3.3 Fluxes of carbon and nitrogen

The fluxes of DOC, DON and NO$_3$-N in runoff were linearly correlated to discharge at both watersheds (Figure 4). The DOC fluxes at the deciduous watershed increased with a much steeper slope in response to discharge than at the mixed, while the NO$_3$-N fluxes at the mixed watershed more steeply increased with increasing discharge than at the deciduous. The POC fluxes were generally much lower than the DOC fluxes, but the POC and PON fluxes increased in a non-linear response to discharge. Only at a single peak flow event on July 14th 2013, the POC fluxes at the mixed watershed were 5 times greater than the DOC fluxes. The same trend was found for the PON and DON fluxes. At the deciduous watershed, only one event caused slightly larger POC than DOC fluxes.

The integrated C and N fluxes over the study period from both watersheds were in the order; DOC > POC and NO$_3$-N > DON > PON (Table 3). The DOC fluxes as the dominant C flux form contributed 75% and 92% of the total organic C flux at the mixed and the deciduous watersheds, respectively. The integrated fluxes of DOC and DON were higher at the deciduous watershed (16 kg C ha$^{-1}$ and 0.5 kg N ha$^{-1}$) than at the mixed watershed (6.7 kg C ha$^{-1}$ and 0.26 kg N ha$^{-1}$). The integrated fluxes of POC and PON were small at both watersheds with only minor differences. Before storm events in June 2013, POC and PON were almost not exported at both watersheds. However, the integrated fluxes of POC and PON increased extremely during heavy storm events in July 2013. The NO$_3$-N fluxes as the dominant N flux form represented 93% and 82% of the total N flux in runoff at the mixed and the deciduous watershed, respectively. The integrated fluxes of NO$_3$-N were about twice as high (5.2 kg N ha$^{-1}$) at the mixed watershed than at the deciduous watershed (2.9 kg N ha$^{-1}$).

### 3.4 Chemical properties of DOM and POM in runoff

The chemical properties of DOM changed with increased discharge at the deciduous watershed, while no significant changes were observed at the mixed watershed (Figure 5). At the deciduous watershed, $SUVA_{280}$ and HIXem increased with increased discharge, while PLF/FLF, PLF/HLF, $\delta^{13}C_{DOC}$ and $\delta^{15}N_{TDN}$ decreased.

At the mixed watershed, the ranges of the DOC/DON ratio, $SUVA_{280}$ and HIXem in runoff were similar to those in throughfall and soil solution, while PLF/FLF and PLF/HLF in runoff corresponded more to those in forest floor percolates (Figure 6). In contrast, at the deciduous watershed, these parameters in runoff were closely related to the quality of forest floor leachates. Also, the $^{13}C$ data in runoff, being more negative at the deciduous watershed, point to a larger proportion of forest floor leachates in runoff than at the coniferous watershed.

The patterns of DOC/DON ratios in response to discharge were also different at the two watersheds (Figure 3c). Large DOC/DON ratios at high discharge at the deciduous watershed resulted from the positive response of DOC concentration and the stable DON concentration to discharge. The DOC/DON ratios at the coniferous watershed were stable in response to discharge.

The range of the POC/PON ratio in runoff was similar to that of the POC/PON ratio of mineral soil layers at both watersheds

(Figure 7). The same holds for the $\delta^{13}C_{POC}$ values. The $\delta^{15}N_{PON}$ in runoff had a huge variation, with averages being larger than those of the forest floor, but less than those of the mineral soil.

### 4 Discussion

#### 4.1 Different response of DOC to increased discharge at the mixed and the deciduous watershed

We intensively sampled 4 heavy rainfall events during the monsoon season, the events representing a substantial proportion

of the annual precipitation in the region. While the number of events was rather small, consistent patters emerged documenting the response of N and C fluxes to precipitation and discharge changes. The increase of DOC concentrations and fluxes in runoff induced by heavy storm events with increased discharge is consistent with the findings of previous studies (Dhillon and Inamdar, 2013; Jeong et al., 2012; Johnson et al., 2006; Lloret et al., 2013). In our study, the response to discharge and the integrated fluxes of DOC in runoff were much larger at the deciduous than at the mixed watershed. Similar

to our results, larger annual DOC fluxes at a deciduous forested catchment than at a mixed coniferous catchment were reported by Amiotte-Suchet et al. (2007).

The different response of DOM in runoff to discharge between the two watersheds, such as the large response of runoff DOC concentration to discharge at the deciduous watershed (Figure 3a) and the significant change in runoff DOC quality parameters (Figure 5), are likely caused by a shift of hydrological flow paths to more surficial layers at the deciduous

watershed. Also, the comparison of DOC quality parameters in runoff with those in forest floor leachates and soil solution at the deciduous watershed (Figure 6) indicated that a larger proportion of the DOC in runoff from forest floor leachates at the

deciduous. Previous studies have also reported a positive relationships between discharge and DOM concentrations in runoff as a consequence of changing hydrologic flow paths from deeper soil to upper soil layers and forest floors at high discharge (Aitkenhead-Peterson et al., 2005; Bass et al., 2011; Sanderman et al., 2009). As several watershed characteristics (slope and soil textures) and the precipitation regime at both watersheds were similar, the differences between the watersheds are likely due to the tree species effects on the infiltration of precipitation water into the soil and on the mobilization of DOM. The tree species effect became obvious although the proportion of coniferous tree species was only 39% of the watershed area. Several processes might be involved to explain the tree species effect: i) In the deciduous litter layer the leaves are overlapping and partly impermeable, which may cause more surface near flow than in coniferous litter layers with relatively large pore spaces in between needles. ii) The relatively higher level of hydrophobicity of coniferous forest floors compared to deciduous forest floors (Butzen et al., 2014) can result in less DOC release from coniferous forest floors. iii) The mobilization of DOC in soils depends on throughfall chemistry (Kalbitz et al., 2000). Throughfall at the MC plot was more acidic (pH 4.7±0.4) and had a higher ionic strength (15.9±11.3 µS cm$^{-1}$) than at the DD plot (pH 6.1±0.2, 10.3±6.3 µS cm$^{-1}$) and the MD plot (pH 5.8±0.4, 9.0±6.3 µS cm$^{-1}$). Acidity and ionic strength are negatively related to DOC release from soils (Clark et al., 2011; Michalzik et al., 2001; Moldan et al., 2012). iv) In stream generation of DOC from litter might be involved (Johnson et al., 2006) if more leaf than needle litter enters the stream. v) As the deciduous watershed is located at a higher altitude the soils might generally be shallower than at the mixed watershed, which will add to the larger near surface flow paths under high precipitation. vi) Faster decomposition of the deciduous litter leaches relatively more DOM resulting in higher DOC export fluxes at the deciduous than at the mixed watershed. Based on our data set of this study, one cannot quantify the relative importance of these factors causing the differences between the watersheds.

## 4.2 Organic and inorganic nitrogen in runoff

At both watersheds, $NO_3$-N was the dominant form of total N flux in runoff (Table 3). Several studies have reported that DON accounted for the dominant fraction of N flux in undisturbed forested watersheds (Alvarez-Cobelas et al., 2008; Frank et al., 2000; Kaushal and Lewis, 2003; Pellerin et al., 2006). Substantial fluxes of $NO_3$-N and the dominance of $NO_3$-N over DON in runoff are likely due to a certain degree of N-saturation (N supply > N demand) of these forested watersheds (Aber et al., 1998; Compton et al., 2003). Hence, the finding of the dominant $NO_3$-N of total N flux implies that the N deposition in the area seems quite high (estimated between 24-51 kg N ha$^{-1}$ yr$^{-1}$; Berger et al., 2013). In July 2013, the integrated N flux in throughfall was however similar at the both watersheds (data not shown). Hence, the differences in N deposition between the two watersheds unlikely explain higher $NO_3$-N fluxes in runoff at the mixed than at the deciduous watershed (Table 3). The C/N ratio of the forest floor was found to be a good indicator of $NO_3$-N release with increasing fluxes at low C/N ratios (Borken and Matzner, 2004; MacDonald et al., 2002). However, the C/N ratio of the organic layer at the mixed watershed (20-28) was higher than at the deciduous watershed (19-21), which does not agree with the findings of MacDonald et al. (2002). Overall, it seems that a larger N uptake by the deciduous trees at the deciduous watershed could explain the differences in the $NO_3$-N fluxes.

**4.3 Particulate organic matter in runoff**

The integrated fluxes of POC and PON during the study period were much less than those of the dissolved elements and did not differ significantly between the watersheds. POC and PON fluxes exceeded their dissolved fractions only for short time during heavy storm events with more than 100 mm precipitation except one storm event at the deciduous watershed on 2013 July 14 (Table 2). Previous studies in the nearby region considered 100 mm precipitation as a threshold that would induce large POC fluxes (Jeong et al., 2012; Jung et al., 2012). Our finding indicates that POM fluxes from forested watershed are unlikely regulated solely by precipitation amount, but slope and river bench characteristics will interfere. The small proportion of particulate fluxes in our study seems to be mainly caused by the relatively moderate precipitation events during the study period. The POC/PON ratios in runoff as well as the $\delta^{13}C_{POC}$ and $\delta^{15}N_{PON}$ were similar to those of the mineral soil and different to those of the forest floor. This indicates that the particulate matter originated from the erosion of mineral soil along the stream benches. Higher annual POC fluxes than DOC fluxes were observed in some mountainous forested watersheds (Kao and Liu, 1997; Kim et al., 2010; Lloret et al., 2013), which does not agree with our finding and some other studies (Dhillon and Inamdar, 2013; Inamdar et al., 2011; Jeong et al., 2012). The differences in findings may be related to the topography of forested watershed because steeper slopes induce higher fluxes of POC (Hilton et al., 2012; Janeau et al., 2014; Jung et al., 2012).

**5 Conclusions**

Our study emphasized the role of heavy precipitation events and vegetation cover for the export fluxes of particulate and dissolved organic C and N with runoff from forested watersheds. Our results suggest that changes of the precipitation regime, with more severe monsoon storms in the future as predicted, will increase the export of dissolved and particulate organic matter from these watersheds. The proportion of coniferous tree species at the mixed watershed was sufficient to induce less DOC fluxes and larger $NO_3$-N fluxes with runoff as compared to the deciduous watershed. Differences in the flow paths between the watersheds are seen as the major trigger for the differences in runoff with a larger proportion of near surface flow at the deciduous watershed. A larger proportion of coniferous forests will likely lead to less inputs of organic carbon and larger inputs of inorganic nitrogen to the receiving surface water bodies.

**Author contribution**

Mi-Hee Lee carried out the experimental work and data evaluation and prepared the manuscript with contribution from all co-authors.

Egbert Matzner and Ji-Hyung Park contributed to the design of this study, to data evaluation, interpretation of results and writing of the manuscript.

Jean-Lionel Payeur-Poirier supported the field work and provided the discharge data.

**Acknowledgements**

This study was accomplished within the framework of the International Research Training Group TERRECO (GRK 1565/1) and funded by the German Research Foundation (Deutsche Forschungsgemeinschaft; DFG) at the University of Bayreuth and the Korean Research Foundation (KRF) at Kangwon National University. We acknowledge the BayCEER Laboratory of Isotope Biogeochemistry for the isotope abundance analysis and the Central Analytical Department of BayCEER for mineral-N measurement at the University of Bayreuth. We are grateful to other TERRECO colleagues for the comprehensive support and to Uwe Hell for the sampler installation of soil solution. We also appreciate the international collaboration with Bomchul Kim, Youngsoon Choi and Jaesung Eum from Kangwon National University (Chuncheon) and with Jin Hur and Bomi Lee from Sejong University (Seoul).

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

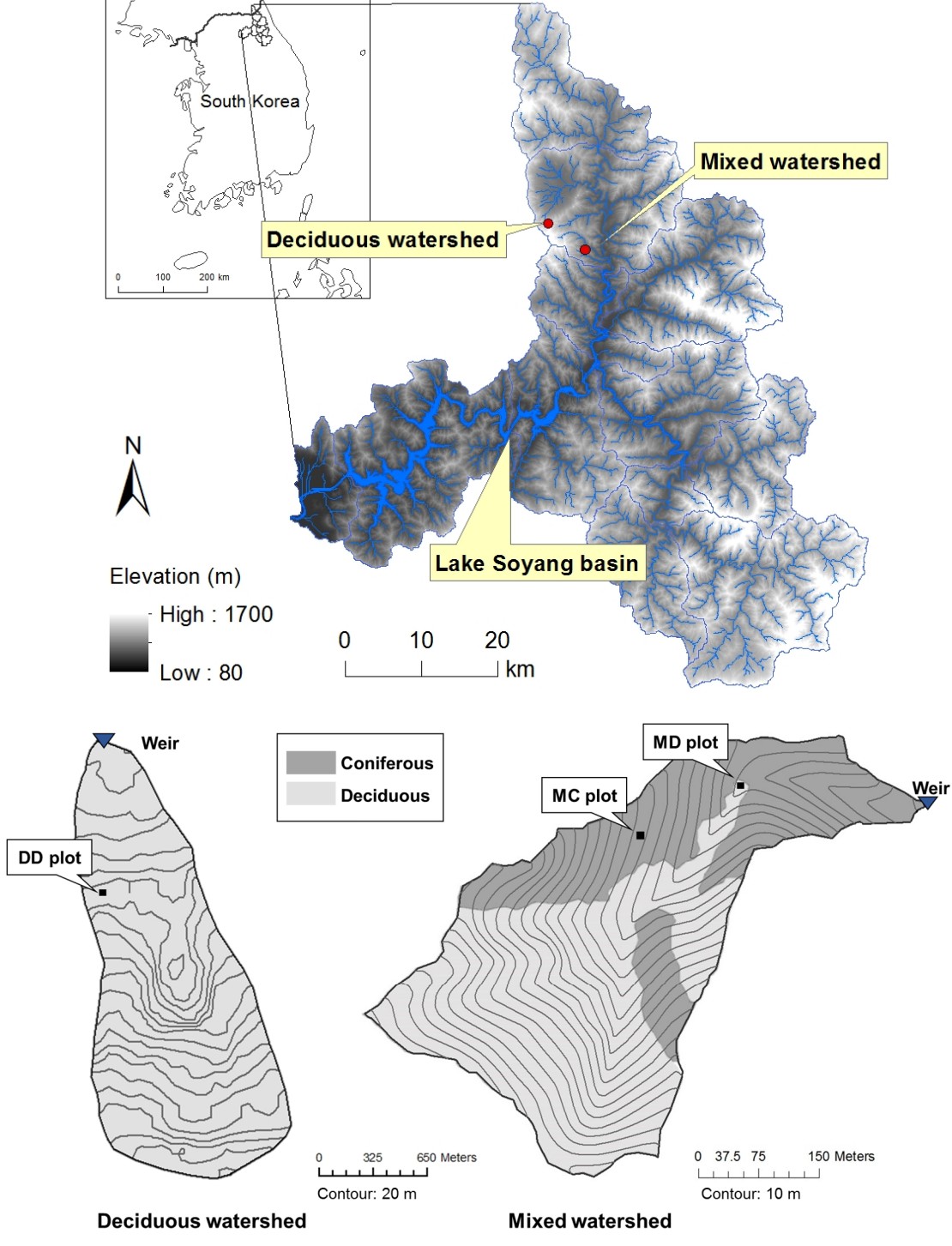

**Figure 1: Location and tree species composition of the two studied forested watersheds. Lake Soyang map was modified from Jung et al. (2015).**

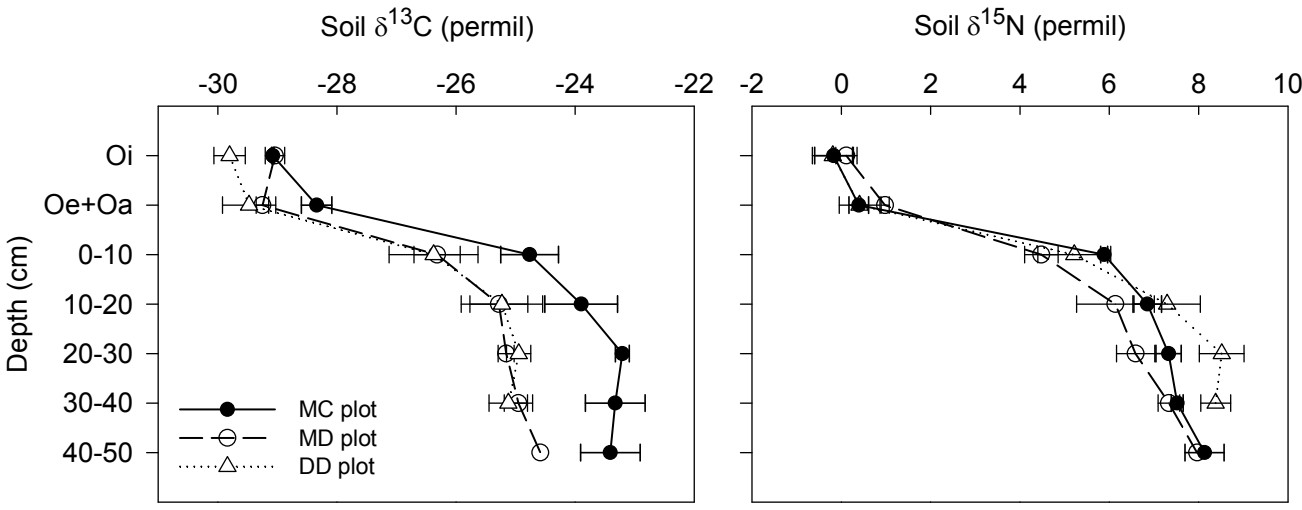

**Figure 2: Soil profiles of** $^{13}$**C and** $^{15}$**N isotope abundance at the MC, MD, and DD plot. Error bars represent standard deviation (n=3).**

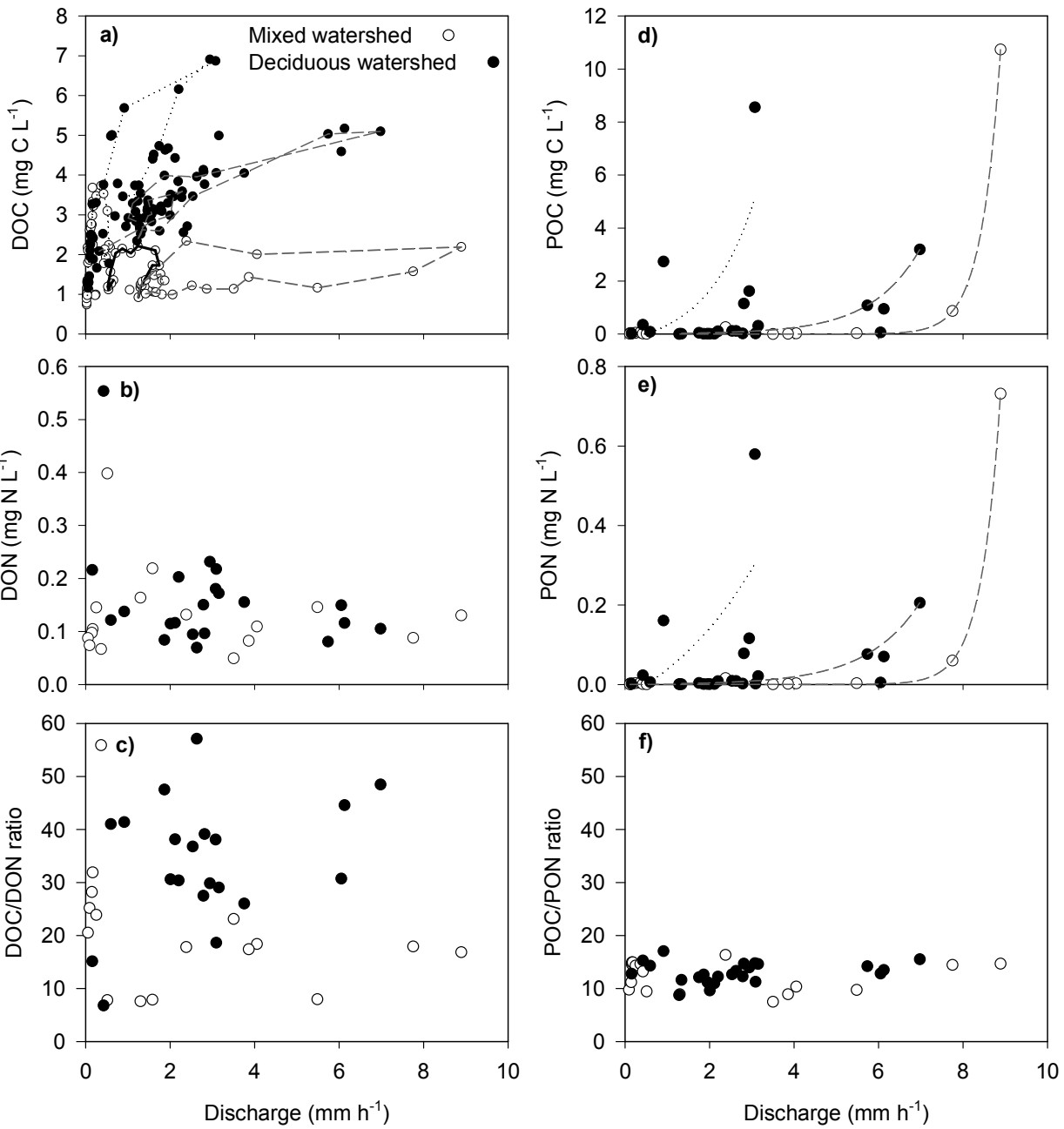

**Figure 3: Concentrations of a) dissolved organic carbon (DOC) and b) nitrogen (DON), d) particulate organic carbon (POC) and e) nitrogen (PON) and the ratios of c) DOC/DON and f) POC/PON in runoff with discharge during monsoon storm events. Doted, solid and dashed lines correspond to the storm event of July 8th 2013, July 11th 2013 and July 14th 2013, respectively.**

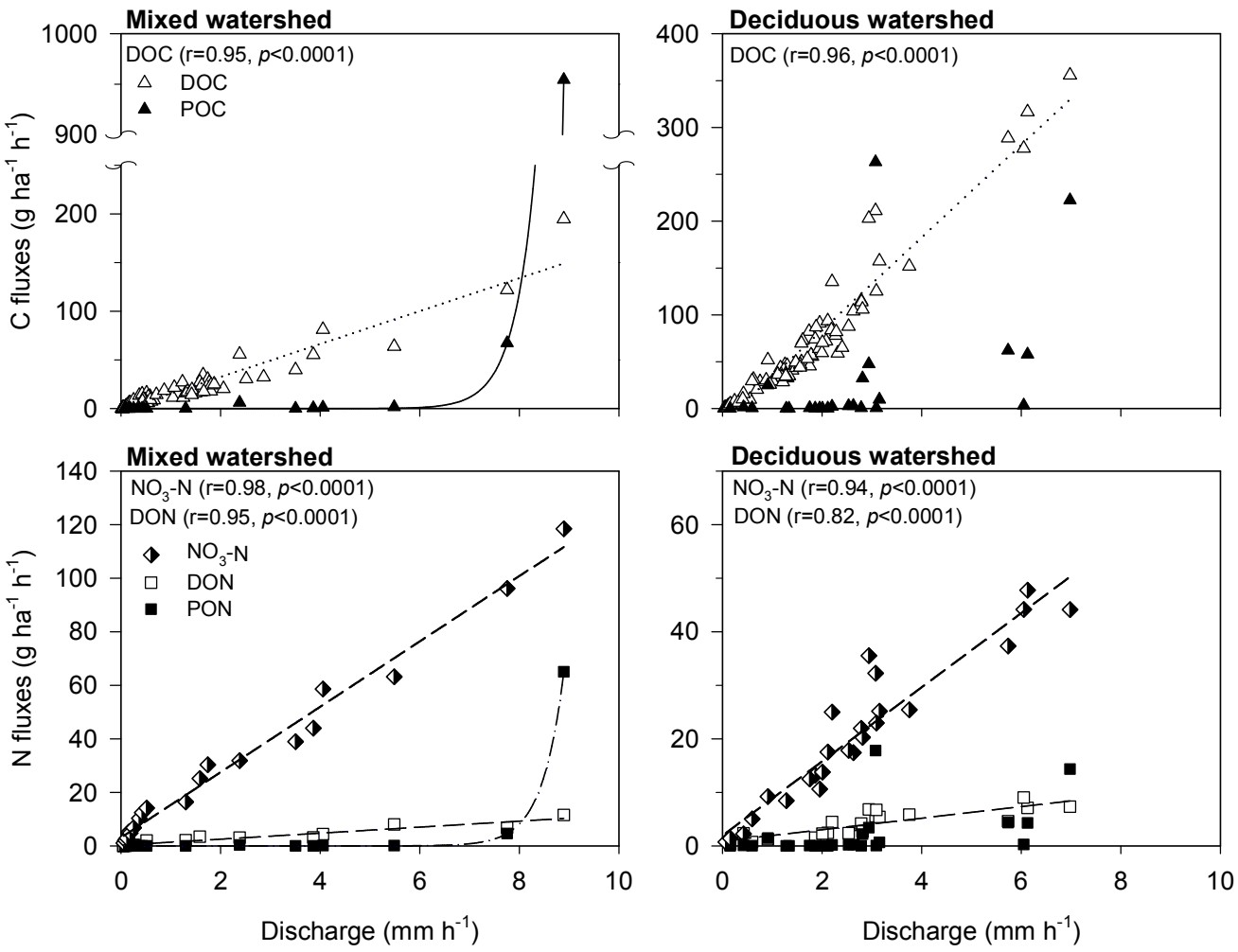

**Figure 4: Fluxes of carbon (dissolved organic carbon (DOC) and particulate organic carbon (POC) and nitrogen (dissolved organic nitrogen (DON), particulate organic nitrogen (PON), and nitrate ($NO_3$-N) in runoff with discharge during monsoon storm events.**

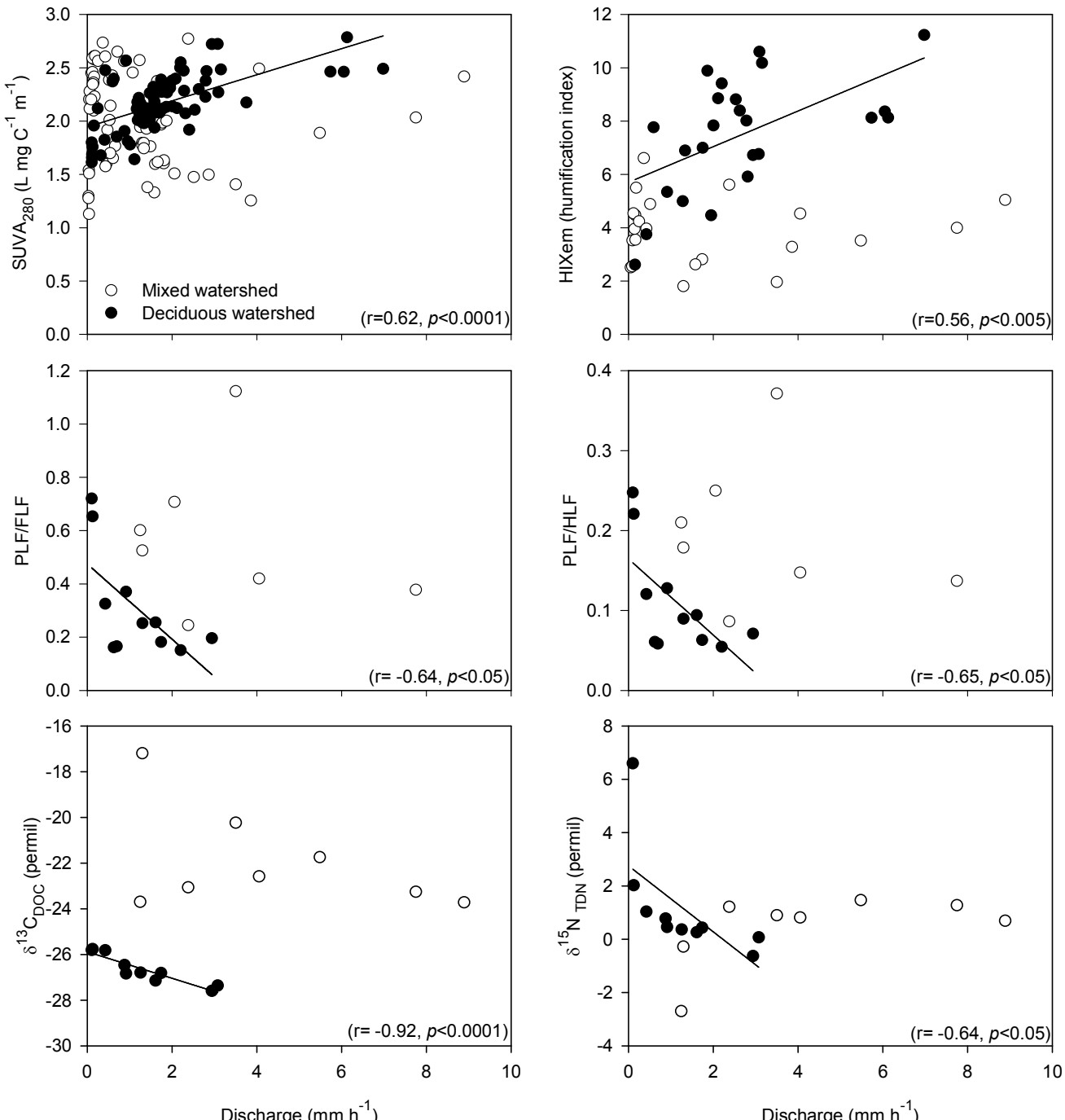

**Figure 5: Specific ultraviolet absorbance (SUVA$_{280}$), humification index (HIXem), protein-like fluorescence/humic-like fluorescence (PLF/HLF), protein-like fluorescence/fulvic-like fluorescence (PLF/FLF), $^{13}$C isotope abundance of dissolved organic carbon ($\delta^{13}C_{DOC}$) and $^{15}$N isotope abundance of total dissolved nitrogen ($\delta^{15}N_{TDN}$) in runoff with discharge during monsoon storm events. Only significant regressions are shown.**

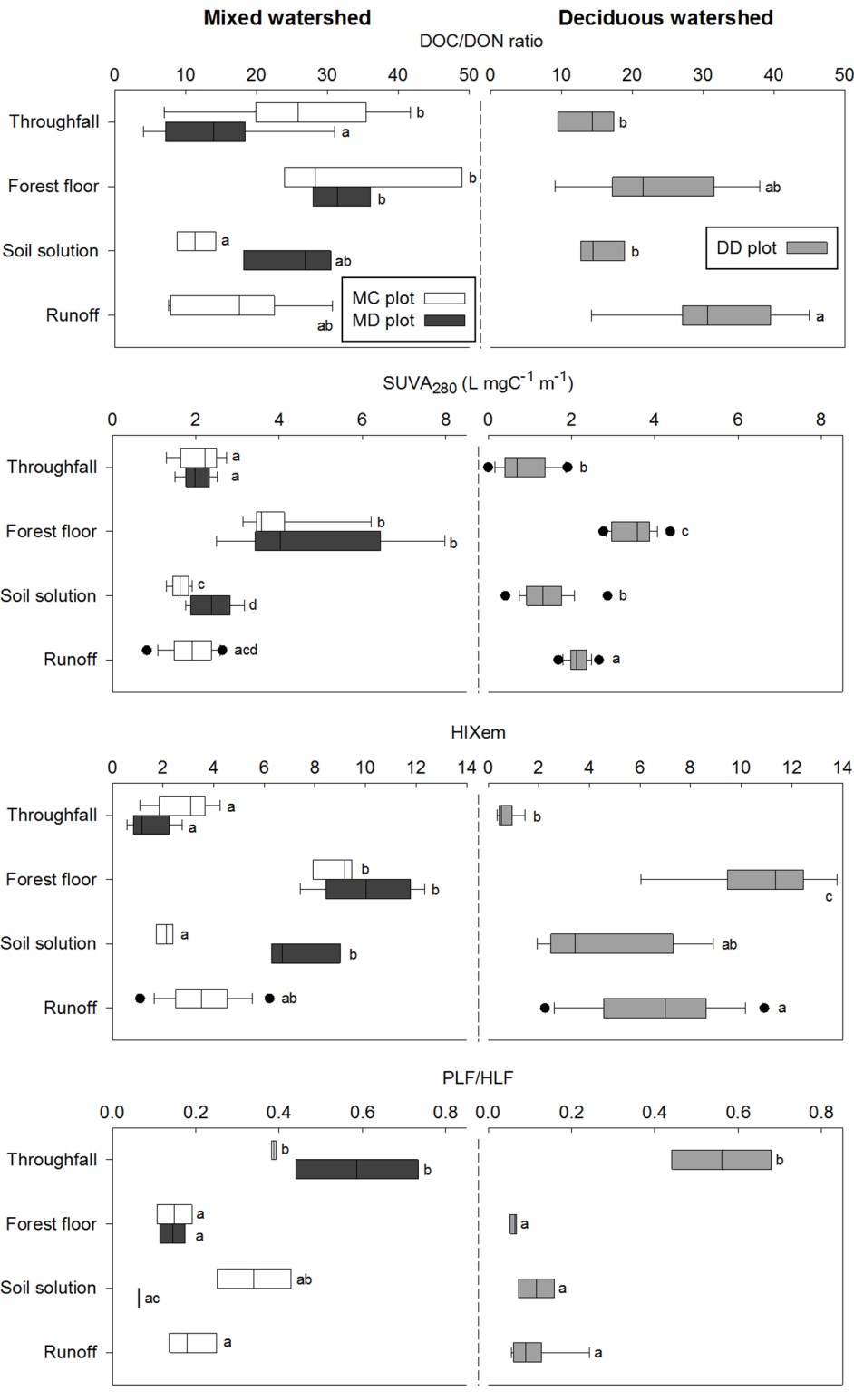

**Figure 6: Range of dissolved organic carbon and nitrogen ratio (DOC/DON ratio), specific ultraviolet absorbance (SUVA$_{280}$), humification index (HIXem), and protein-like fluorescence/humic-like fluorescence (PLF/HLF) of throughfall, forest floor leachates, soil solution, and runoff during monsoon storm events. Box plots display minimum, lower quartile, median, upper quartile, maximum and outliers. Statistically significant differences between sample types (throughfall, forest floor leachates, soil solution, and runoff) are indicated by different letters in the box plots, significance level of $p<0.05$.**

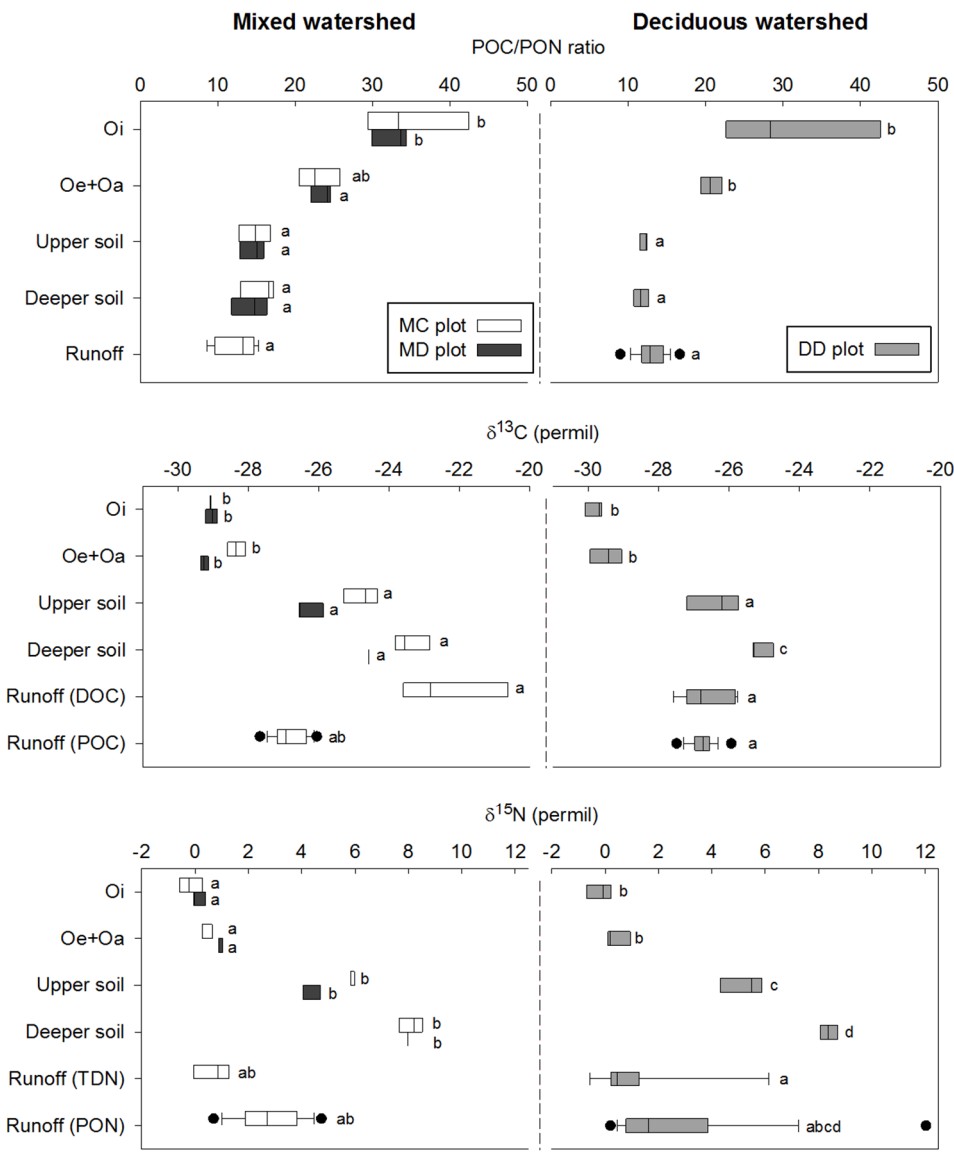

**Figure 7: Range of particulate organic carbon and nitrogen ratio (POC/PON ratio), δ¹³C and δ¹⁵N in Oi, Oe+Oa, upper soil (0-10 cm depth), deeper soil (40-50 cm depth at the MC and MD plot, 30-40 cm depth at the DD plot), and runoff. Box plots display minimum, lower quartile, median, upper quartile, maximum and outliers. Statistically significant differences between sample types (Oi, Oe+Oa, upper soil, deeper soil, and runoff) are indicated by different letters in the box plots, significance level of *p<0.05*.**

**Table 1: Tree species composition and geomorphological characteristics of the studied forested watersheds.**

| Watershed | Major tree species | Area | Average Slope | Altitude |
|---|---|---|---|---|
| | | (ha) | (°) | (m a.s.l.) |
| Mixed | | (Total 15.6) | 27.9 | 368-682 |
| Coniferous | Larch and Pine | 6.1 | | |
| Deciduous | Walnut, Maple, Oak, Lime, Elm | 9.5 | | |
| Deciduous | Walnut, Maple, Oak, Ash | 39 | 24.0 | 586-1005 |

a.s.l.: above sea level

**Table 2: Hydrological characteristics of sampled storm events and maximum concentration of dissolved organic carbon (DOC) and nitrogen (DON), particulate organic carbon (POC) and nitrogen (PON) in runoff.**

| Watershed | Start time | Duration | Number of samples | Total precipitation | Maximum precipitation intensity | Average precipitation intensity | Maximum discharge | Discharge before start of a storm event | Maximum DOC | Maximum DON | Maximum POC | Maximum PON |
|---|---|---|---|---|---|---|---|---|---|---|---|---|
| | | (h) | | (mm) | (mm h$^{-1}$) | (mm h$^{-1}$) | (mm h$^{-1}$) | (mm h$^{-1}$) | (mg C L$^{-1}$) | (mg N L$^{-1}$) | (mg C L$^{-1}$) | (mg N L$^{-1}$) |
| Mixed | 2013 July 02  9:00 | 15 | 16 | 40.0 | 8.5 | 2.7 | 0.17 | 0.03 | 3.7 | 0.1 | 0.04 | 0.002 |
| | 2013 July 08  3:00 | 24 | 15 | 56.5 | 10.0 | 2.3 | 0.55 | 0.04 | 3.7 | 0.4 | 0.06 | 0.004 |
| | 2013 July 11  9:00 | 12 | 12 | 44.5 | 10.0 | 3.7 | 1.47 | 0.52 | 2.1 | 0.2 | 0.03 | 0.003 |
| | 2013 July 14  2:00 | 41 | 26 | 172.5 | 34.0 | 4.2 | 8.89 | 1.21 | 2.4 | 0.2 | 10.7 | 0.730 |
| | | | | (total 313.5) | (avg. 16) | (avg. 3.2) | (avg.2.8) | (avg. 0.45) | | | | |
| Deciduous | 2013 July 08  3:00 | 32 | 21 | 117.5 | 20.0 | 3.6 | 3.16 | 0.10 | 6.9 | 0.6 | 8.6 | 0.58 |
| | 2013 July 11  9:00 | 15 | 20 | 43.5 | 8.0 | 2.9 | 3.07 | 0.58 | 5.0 | 0.2 | 0.3 | 0.02 |
| | 2013 July 14  2:00 | 42 | 23 | 148.5 | 32.0 | 3.5 | 7.39 | 1.07 | 5.1 | 0.2 | 3.2 | 0.21 |
| | 2013 July 18  14:00 | 9 | 10 | 58.0 | 20.5 | 6.4 | 6.61 | 0.32 | 5.2 | 0.2 | 1.1 | 0.08 |
| | | | | (total 367.5) | (avg. 20) | (avg. 4.1) | (avg. 5.1) | (avg. 0.52) | | | | |

**Table 3: Total precipitation, total runoff and integrated fluxes of dissolved organic carbon (DOC) and nitrogen (DON), nitrate (NO$_3$-N), particulate organic carbon (POC) and nitrogen (PON) in June and July 2013.**

| Watershed | Period | Total precipitation (mm) | Total runoff (mm) | DOC fluxes (kg C ha$^{-1}$) | DON fluxes (kg N ha$^{-1}$) | NO$_3$-N fluxes (kg N ha$^{-1}$) | POC fluxes (kg C ha$^{-1}$) | PON fluxes (kg N ha$^{-1}$) |
|---|---|---|---|---|---|---|---|---|
| Mixed | June[a] | 86.0 | 21.8 | 0.22 | 0.02 | 0.43 | 0.001 | 0.0001 |
| | July[b] | 508.0 | 380.7 | 6.74 | 0.26 | 5.20 | 2.22 | 0.15 |
| Deciduous | June[a] | 70.5 | 52.4 | 0.85 | 0.1 | 0.52 | 0.01 | 0.001 |
| | July[b] | 498.0 | 439.5 | 16.13 | 0.52 | 2.87 | 1.46 | 0.11 |

[a] Before heavy storm events from June 1st to June 30th, 2013

[b] Heavy storm events from July 1st to July 20th, 2013