# Peer review of "Variability in runoff fluxes of dissolved and particulate carbon and nitrogen from two watersheds of different tree species during intense storm events"

_Biogeosciences, 2016_

## Referee Comment (RC1) · D. Kumar (Referee) · 31 May 2016

General comments The manuscript presented very useful information on forms and flows of carbon and nitrogen in an upland fluvial system. The planning and coverage of sampling and parameters are good. The results will improve our understanding of material flows in terrestrial fluvial systems based on nature of tree types and water flows. The results deserve to be published but not in the present form. The basic problem is with the presentation and the way they dealt with scientific and technical issues. I recommend encouraging the authors for submitting a revised version after

they work on some of the issues mentioned and improve the presentation for clarity.

Specific Comments Title: manuscript is not really studying the 'response of carbon and nitrogen components IN RUNOFF to storm events' but addresses the influence of storms on carbon and nitrogen components in runoff. The appropriate title for the manuscript seems to be (from lines 32-33 on page 2): Influence of tree species and episodic discharges on the fluxes of dissolved and particulate carbon and nitrogen from two watersheds OR Changes in fluxes of dissolved and particulate carbon and nitrogen from two watersheds of different tree types during heavy discharge periods.

Page 3: Lines 4-5 – The sentence 'the annual air temperature ranges from 10-15oC with -6oC in January and 26oC in August' does not make sense to me.

Page 3: Line 7 –Is the 47% broadleaved forest 'the deciduous'? Here the comparison is between deciduous and mixed types and so appropriate type to be named than the description (broadleaved).

Page 3: Lines 12-14 – Are the slopes at two sampling points MC and MD oriented in different directions in the mixed watershed? If they are oriented in one direction then fluxes from the upper can definitely influence the other during floods. This question also pertains to slope comparisons between two watershed sampling points. Figure 1 and info on page 3 shows that deciduous sampling point is at a higher altitude than that in the mixed watershed. What if both the watersheds (and hence the sampling points) slope in the same direction? If yes, the flow from deciduous sampling point would influence the composition at MC and MD!! This is quite possible as the two watersheds are nearby. The authors should clarify on this issue of slopes and possible interference between sampling points.

Page 4: Line 5 – 'were collected after each storm event': maximal flows/fluxes must have occurred during the peak flow. When the maximal speeds subsided the original peak signals (of concentrations/fluxes) of the flood may have been lost!! This can be exemplified using the data in Table 2 for deciduous station. On July 8 (say first flood

studied) DOC, DON, POC, PON values are higher than the following flood event on 14 July. Obviously, the first flood water carried more C & N than the second one since the first/fresh rain/flood can dissolve/scoop more of materials accumulated during the preceding dry or intervening periods in the soils. This was also noticed by the authors on Page 6 Line 15.

Page 4: Line 5 – 'runoff samples were collected every 1 or 2 h in the weir' – were these also collected after the storm events (coinciding with throughfall, forest floor leachate and soil solution sampling) or during the event or both? This information is crucial for making the right comparisons and assessing changes.

Page 6: line 3 – In the absence of clear definitions of Oi, Oe and Oa it is hard to understand the significance of percentages of these fractions, as also in relevant Figures.

Page 6: Lines 17-21 – I am not convinced of the 'threshold value' since there are not enough data points to show a consistent increase in concentrations. Relatively higher concentrations in POC and PON are found (Fig 2d,e) RANDOMLY during discharges from ∼1 to 9 mm/h.

Page 6: Lines 22-23 – DOC rise with increasing flood and fall with decreasing flood is convincing and is, indeed a good observation.

Page 8: in the entire page of this discussion, the authors did not seem to have paid much attention to (a) nature of litter, (b) altitude and (c) substratum of the two watersheds. I understood that the deciduous watershed was at higher altitude with hard rock below 40 cm whereas that of mixed was at lower height laden with soils upto or below 50 cm. Presumably the hard rock might have occurred deeper in the mixed watershed. The nature off litter (seasonally fallen parts of the trees) would be relatively freshly fallen in the deciduous watershed that could easily be broken/decomposed by physical/microbial activities that could leach more DOC or dissolvable OM. This fresh DOM can be easily be flushed by flowing water. The hard substratum and high altitude facilitate flow of water at higher speeds (as it cannot seep deep) in the deciduous watershed than in the mixed watershed. The rapidly draining flood facilitates easy mixing of forest floor and soil solutions with surface runoff.

Although logical in their statement of "i) In the deciduous litter layer the leaves are overlapping and are partly impermeable which may cause more surface near flow (lines 23-25)" this does support their observation that 'a larger proportion of the DOC in runoff results from forest floor leachates at the deciduous (lines 17-18)'. If the top layer is impermeable how would one explain high DOC in runoff to have come from mixing with forest floor leachates? It is also possible for high DOC formation at the surface itself as the fresher litter is weathered or decomposed on the floor of the deciduous watershed.

Page 9: Lines 2-8 – higher DOC/DON ratio at deciduous basin is possible when organic matter with less or no nitrogen but rich in carbon is weathered and is leached. Nitrogen compounds are perhaps enriched in litter or particulates. However, Fig 2 depicts lower PON than DON, in general in both watersheds, implying that nitrogen might be remaining with the deposited litter in the watersheds!

Page 9: Statements in lines 11-13 ('Substantial fluxes of NO3-N and the dominance of NO3-N over DON in runoff are likely due to a certain degree of N-saturation (N supply > N demand) of these forested watersheds (Aber et al., 1998; Compton et al., 2003)') and lines 20-21 ('Overall, it seems that a larger N uptake by the deciduous trees at the deciduous watershed could explain the differences in NO3-N fluxes') are arbitrary and not supported by any data.

Technical Comments Page 3: Lines 11 and 21 – Are the latitude and longitude positions accurate to the decimals mentioned?

Page 3: Line 30: Does 'throughfall' refer to precipitation or rainwater?

Page 4: Line 13 – define Oi, Oe and Oa.

Page 4: Lines 26-27 – 'The storm events during monsoon season were identified from

the start to the end of precipitation with more than a day interval between each storm event'. The storm events should be identified based on meteorological observations of wind and rainfall. However, one should keep in mind that the present study is made in summer monsoon. During monsoon season the rainfall may not be continuous on all days but with intermittent gaps (breaks) or spells of rain. I guess the authors are referring to these spells, or at the most the episodic rainfall events (which are normal during summer monsoon) of variable duration as 'storms'. This requires authors clarification for what they meant by 'storm'. This point, however, was rightly stated by the authors on page 8 line 4 – 'four heavy rainfall events of the monsoon season at both watersheds' but not elsewhere in the manuscript.

Page 5: Lines 4 and 8 – (i) unmatched DOC and POC cutoff limits! (ii) 0.7 micron cutoff limit for POC is quite on higher side since most of the fine sized particulate materials are lost through the filter paper.

Page 5: Line 6 – was nitrite in water analysed? It should be included in mineral-N.

Fig 2 – what are FPOC/FPON in Fig. 2f?

Fig. 3. Upper panel in the left column – DON and PON should be corrected to DOC and POC.

Figures 5 & 6: Alphabets (a, b....) need to be explained in more detail. For instance what does it mean by ab or abcd. In the captions it is mentioned "Different alphabet letters indicate the significant difference between groups". I could not understand what is the difference and what are the groups mentioned.

Fig. 6 – DTN is dissolved total nitrogen?

---

## Referee Comment (RC2) · Anonymous Referee #1 · 26 Jun 2016

Lee et al. in this manuscript have tried to understand the effect of storm events on dissolved and particulate carbon and nitrogen in runoff from two different watersheds dominated with different species of vegetation. In general, study of this kind can provide an improved understanding of nutrients and material transfer from different terrestrial set up under monsoon condition. It is understandable that it requires a lot of effort to carry out study of this nature; it would have been better if there were more sampling events. Keeping aside the limitation in numbers of sampling events, the manuscript in present form is very poorly written with lots of mistakes in presentation of results and figures. I believe it is not suitable for publication in Biogeosciences. Below are some of

my specific observations:

* Abstract does not clearly bring out the findings of the study. It merely describes the variability in results. I think they should modify it to include the processes involved for such observation.

* Introduction is poorly written with no focus. It wanders from one topic to other other without gravitating towards the focus of the work. The first line of the introduction itself appears confusing to me.

* One common observation throughout the manuscripts is regarding the references. I think it should be chronologically arranged.

* Framing of the sentences from previous works is such that they appear as if they are from the present study (line 15-20).

* Introduction last sentence: What is measuring campaign?

* How can annual air temperature range from 10 - 15 oC with -6oC in January and 26 oC in August?

* Page 3: Line 20: '(deciduous watershed) (Figure 1)' should be replace with (deciduous watershed; Figure 1).

Page 4: The text suddenly jumps to Oi and Oe+Oa layers without providing any context to it.

Page 4: What do you mean by partly below detection limit. Please provide the number of samples or occasions when it went below detection limit.

* The authors mention the statistical methods followed for analysis but it hardly comes during discussion.

* Page 5: Line 25: Elaborate on the meaning of freeze drying the samples for mass spectrometric analysis or rephrase the sentence.

* Should not mineral N include nitrite? * Authors state that 'the soil $\delta$13C and soil $\delta$15N values significantly increased with soil depth from -29 to -24‰ and from 0 to 8‰ respectively". However, it would be nice to see the vertical profiles of such data. Was the surface $\delta$15N always near 0 ‰

* I could not make the sense out of the sentence 'In the study period, the highest precipitation coincided with the maximum precipitation intensity, the highest precipitation intensity and the maximum discharge at the 10 mixed watershed and at the deciduous watershed on July 14th, 2013"

* While the difference in DOC concentrations with discharge between deciduous and mixed watershed appears to be convincing (Fig 2a), the POC and PON increase with discharge relies heavily on one data point from high discharge. I do not doubt the increase but I believe that to make unequivocal conclusion more points would have been an asset.

* Please rephrase the sentence "The fluxes of DOC and NO3-N increased with a much steeper slope at the deciduous and at the mixed watershed, respectively".

* There is problem with the symbols and its representation in Figure 3. I think authors should be careful with these kinds of mistakes before submitting their manuscript for review. It is very tiring to review a manuscript with these kinds of mistakes.

* Result section discussing Chemical properties of DOM and POM in runoff should be modified with proper emphasis on isotopic data. At present the isotopic data has just been mentioned as passing comment.

* In the discussion, authors have admitted that the numbers of events are rather low in the study and observations made by them have already been observed before by Dhillon and Inamdar (2013). I am wondering what novel finding they are discussing which warrants publication in a journal like Biogeosciences.

* Discussion section needs to be re-written with proper emphasis on the major findings

from this work. The mechanisms and processes behind the differences in observation need to be discussed properly. The effect of altitude, nature of littler and specific nature of the two watersheds needs to be take n in account.

\*Fig 2: What are FPOC/FPON?

Fig 5: A succinct Fig 5 will be better.

---

## Author Comment (AC1) · 22 Jul 2016

General comments
The manuscript presented very useful information on forms and
flows of carbon and nitrogen in an upland fluvial system. The planning and coverage
of sampling and parameters are good. The results will improve our understanding of
material flows in terrestrial fluvial systems based on nature of tree types and water
flows. The results deserve to be published but not in the present form. The basic
problem is with the presentation and the way they dealt with scientific and technical
issues. I recommend encouraging the authors for submitting a revised version after they work on some
of the issues mentioned and improve the presentation for clarity.

*Reply We appreciate your valuable comments and suggestions to improve our manuscript. We will
revise our manuscript incorporating all your comments.*

Specific Comments
Title: manuscript is not really studying the 'response of carbon
and nitrogen components IN RUNOFF to storm events' but addresses the influence
of storms on carbon and nitrogen components in runoff. The appropriate title for the
manuscript seems to be (from lines 32-33 on page 2): Influence of tree species and
episodic discharges on the fluxes of dissolved and particulate carbon and nitrogen from
two watersheds OR Changes in fluxes of dissolved and particulate carbon and nitrogen
from two watersheds of different tree types during heavy discharge periods.

*Reply title will be changed accordingly to 'Changes in fluxes of dissolved and particulate carbon and
nitrogen from two watersheds of different tree species during intense storm events'.*

Page 3: Lines 4-5 – The sentence 'the annual air temperature ranges from 10-15oC
with -6oC in January and 26oC in August' does not make sense to me.

*Reply Clarified as    'The average annual temperature of the Gangwon-province is $11^oC$ with monthly
average temperature ranging from $-5^oC$ in January to $24^oC$ in August during from 1981 to 2010'.*

Page 3: Line 7 –Is the 47% broadleaved forest 'the deciduous'? Here the comparison
is between deciduous and mixed types and so appropriate type to be named than the
description (broadleaved).

*Reply clarified as    'Korean mountainous forests are mostly composed of deciduous forests
representing 47% of the total forested area (38% coniferous forest, 12% mixed deciduous and
coniferous forest)'.*

Page 3: Lines 12-14 – Are the slopes at two sampling points MC and MD oriented in
different directions in the mixed watershed? If they are oriented in one direction then
fluxes from the upper can definitely influence the other during floods. This question
also pertains to slope comparisons between two watershed sampling points. Figure 1
and info on page 3 shows that deciduous sampling point is at a higher altitude than that
in the mixed watershed. What if both the watersheds (and hence the sampling points)
slope in the same direction? If yes, the flow from deciduous sampling point would
influence the composition at MC and MD!! This is quite possible as the two watersheds
are nearby. The authors should clarify on this issue of slopes and possible interference
between sampling points.

*Reply The slope direction of the coniferous part of the mixed watershed is towards the MD plot.
Lateral flow from coniferous part to MD can only influence deeper soil solution characteristics at MD
as near surface flow was never observed. Data from Fig. 5 indicate significant quality differences of
soil solutions between the MD and MC plots which suggest only a minor influence on soil solution
chemistry at MD from lateral flows. Moreover, the quality parameters of soil solutions at the MD plot
are similar to those of the DD plot, the latter being not influenced by lateral flows from coniferous
sites. Thus, we believe that the components at the MC plot did not directly affect those at the MD plot.*

Page 4: Line 5 – 'were collected after each storm event': maximal flows/fluxes must
have occurred during the peak flow. When the maximal speeds subsided the original

peak signals (of concentrations/fluxes) of the flood may have been lost!! This can be exemplified using the data in Table 2 for deciduous station. On July 8 (say first flood studied) DOC, DON, POC, PON values are higher than the following flood event on 14 July. Obviously, the first flood water carried more C & N than the second one since the first/fresh rain/flood can dissolve/scoop more of materials accumulated during the preceding dry or intervening periods in the soils. This was also noticed by the authors on Page 6 Line 15.

*Reply: We will clarify this: new text: 'During storm events in July 2013, throughfall, forest floor leachate and soil solution were collected after each storm event so that these samples represent cumulative water samples from corresponding compartments during the entire storm event".*

Page 4: Line 5 – 'runoff samples were collected every 1 or 2 h in the weir' – were these also collected after the storm events (coinciding with throughfall, forest floor leachate and soil solution sampling) or during the event or both? This information is crucial for making the right comparisons and assessing changes.

*Reply technical comment: can easily be clarified as 'Runoff samples were collected at the weir using automatic collectors before, during, and after each rain event at the two watershed, especially during event at intervals of 1 or 2 h during'.*

Page 6: line 3 – In the absence of clear definitions of Oi, Oe and Oa it is hard to understand the significance of percentages of these fractions, as also in relevant Figures.

*Reply definition and thickness of horizons will be given: new text: 'The total stock of organic horizons (Oi: slightly decomposed recognizable litter, Oe: moderately decomposed fragmented litter, Oa: highly decomposed humus material) was collected at each plot in a 20 × 20 cm frame with 10 replicates. The average thickness of Oi and Oe+Oa was 1.2 and 1.5 cm at the MC plot, 2.5 and 3 cm at the MD plot, and 2.3 and 2 cm at the DD plot, respectively'.*

Page 6: Lines 17-21 – I am not convinced of the 'threshold value' since there are not enough data points to show a consistent increase in concentrations. Relatively higher concentrations in POC and PON are found (Fig 2d,e) RANDOMLY during discharges from _1 to 9 mm/h.

*Reply the reviewer is right, text will be changed accordingly to 'At discharges from ~1 to 9 mm h$^{-1}$, higher concentrations of POC and PON in runoff were found (Figure 2d,e). For example, The POC concentration in streamwater from the mixed watershed was as high as 10.7 mg C L$^{-1}$ at the largest discharge of 9 mm h$^{-1}$. At the deciduous watershed, POC concentration reached a maximum of 8.6 mg C L$^{-1}$ already at 3 mm h$^{-1}$ discharge during the first storm event (Figure 2d, Table 2).*

Page 6: Lines 22-23 – DOC rise with increasing flood and fall with decreasing flood is convincing and is, indeed a good observation.

*Reply thank you for your comment; no reply needed.*

Page 8: in the entire page of this discussion, the authors did not seem to have paid much attention to (a) nature of litter, (b) altitude and (c) substratum of the two watersheds. I understood that the deciduous watershed was at higher altitude with hard rock below 40 cm whereas that of mixed was at lower height laden with soils upto or below 50 cm. Presumably the hard rock might have occurred deeper in the mixed watershed. The nature off litter (seasonally fallen parts of the trees) would be relatively freshly fallen in the deciduous watershed that could easily be broken/decomposed by physical/microbial activities that could leach more DOC or dissolvable OM. This fresh DOM can be easily be flushed by flowing water. The hard substratum and high altitude facilitate flow of water at higher speeds (as it cannot seep deep) in the deciduous wa-tershed than in the mixed watershed. The rapidly draining flood facilitates easy mixing of forest floor and soil solutions with surface runoff.
Although logical in their statement of "i) In the deciduous litter layer the leaves are overlapping and are partly impermeable which may cause more surface near flow (lines 23-25)" this does support their observation that 'a larger proportion of the DOC in runoff

results from forest floor leachates at the deciduous (lines 17-18)'. If the top layer is impermeable how would one explain high DOC in runoff to have come from mixing with forest floor leachates? It is also possible for high DOC formation at the surface itself as the fresher litter is weathered or decomposed on the floor of the deciduous watershed.

*Reply: We agree with the last statement and our argumentation in the discussion can be modified for more clarity. Your comments will be incorporated in more detailed discussion. New: 'The deciduous watershed is located at higher altitude suggesting more shallow soils than at the mixed watershed. This may explain the larger near surface flow paths at the deciduous watershed. Moreover, faster decomposition of the deciduous litter leaches relatively more DOM and both factors result in higher DOC export fluxes at the deciduous watershed than at the mixed watershed'.*

Page 9: Lines 2-8 – higher DOC/DON ratio at deciduous basin is possible when organic matter with less or no nitrogen but rich in carbon is weathered and is leached. Nitrogen compounds are perhaps enriched in litter or particulates. However, Fig 2 depicts lower PON than DON, in general in both watersheds, implying that nitrogen might be remaining with the deposited litter in the watersheds!

*Reply we explained the larger DOC/DON ratios at the deciduous watershed in lines 2-8. The mobilization of particulate organic matter is attributed to the erosion or river benches and no conclusion on N retention in litter is possible.*

Page 9: Statements in lines 11-13 ('Substantial fluxes of NO3-N and the dominance of NO3-N over DON in runoff are likely due to a certain degree of N-saturation (N supply > N demand) of these forested watersheds (Aber et al., 1998; Compton et al., 2003)') and lines 20-21 ('Overall, it seems that a larger N uptake by the deciduous trees at the deciduous watershed could explain the differences in NO3-N fluxes') are arbitrary and not supported by any data.

*Reply: true, the conclusion is speculative and that is why it is formulated as a suggestion. However, other reasons for the higher $NO_3$ fluxes at the coniferous site are not likely.*

Technical Comments Page 3: Lines 11 and 21 – Are the latitude and longitude positions accurate to the decimals mentioned?

*Reply technical comment: can easily be clarified. The positions of watershed at the weir will be corrected as 38°12′24.8″N, 128°11′9.1″E for the mixed watershed in Seohwa and 38°15′5.6″N, 128°7′10.9″E for the deciduous watershed in Haean.*

Page 3: Line 30: Does 'throughfall' refer to precipitation or rainwater?

*Reply The sentence will be written as 'Throughfall collectors (n=5) under the canopy were equipped with filters to prevent large particles from entering'.*

Page 4: Line 13 – define Oi, Oe and Oa.

*Reply As mentioned earlier, the definitions of Oi, Oe and Oa will be added in Page 4: line 13*

Page 4: Lines 26-27 – 'The storm events during monsoon season were identified from the start to the end of precipitation with more than a day interval between each storm event'. The storm events should be identified based on meteorological observations of wind and rainfall. However, one should keep in mind that the present study is made in summer monsoon. During monsoon season the rainfall may not be continuous on all days but with intermittent gaps (breaks) or spells of rain. I guess the authors are referring to these spells, or at the most the episodic rainfall events (which are normal during summer monsoon) of variable duration as 'storms'. This requires authors clarification for what they meant by 'storm'. This point, however, was rightly stated by the authors on page 8 line 4 – 'four heavy rainfall events of the monsoon season at both watersheds' but not elsewhere in the manuscript.

*Reply As you suggested, the rainfall characteristic during monsoon will be referred as 'During monsoon season the rainfall was not continuous on all days but with intermittent gaps of rain, thus the*

*most lasting rainfall events were identified as storm events with more than a day interval between each storm event`*
*. The term 'rainfall' will be used to explain 'storms' in the method part of this manuscript.*

Page 5: Lines 4 and 8 – (i) unmatched DOC and POC cutoff limits! (ii) 0.7 micron cutoff limit for POC is quite on higher side since most of the fine sized particulate materials are lost through the filter paper.

**Reply** *We do not agree that the fraction from 0.45 to 0.7 micron represents most of the fine sized material. DOM is commonly defined as organic matter in water samples smaller than 0.45µm (Thurman, 1985). Previous studies have often used a 0.7 µm pore size of glass filter for POM fraction for technical aspects in the analysis (Bauer and Bianchi 2011, Mostofa et al. 2013). Consequently, DOC and POC cutoff limits are unmatched as you pointed. However, prior tests (Doyle 2013) showed that materials between 0.45 and 0.7 µm comprised a minor fraction in total organic matter. We will make a comment on that in the methods section*

> *-Thurman, E. M. (1985). Organic geochemistry of natural waters. Nordrecht, The Netherlands: Martinus Nijhoff/Junk Publisher.*
> *-Bauer, J.E., and Bianchi, T.S. (2011). 5.02—dissolved organic carbon cycling and transformation. Treatise on estuarine and coastal science, 5, 7-67.*
> *-Mostofa, K.M., Liu, C.Q., Minakata, D., Wu, F., Vione, D., Mottaleb, M.A., ... and Sakugawa, H. (2013). Photoinduced and Microbial Degradation of Dissolved Organic Matter in Natural Waters. In Photobiogeochemistry of Organic Matter. Springer Berlin Heidelberg, 273-364*
> *-Doyle, C. B. (2013). Contribution of bacterial cells to the fluorescence spectra of natural organic matter in freshwaters, University of North Carolina, master thesis.*

Page 5: Line 6 – was nitrite in water analysed? It should be included in mineral-N.

**Reply** *Nitrite was not measured because it was negligible in soil solutions and runoff from test measuremenst.*

Fig 2 – what are FPOC/FPON in Fig. 2f?

**Reply** *technical comment: can easily be clarified. Will be changed to POC/PON.*

Fig. 3. Upper panel in the left column – DON and PON should be corrected to DOC and POC.

**Reply** *technical comment: figure will be corrected. Will be corrected to DOC and POC.*

Figures 5 & 6: Alphabets (a, b: : :.) need to be explained in more detail. For instance what does it mean by ab or abcd. In the captions it is mentioned "Different alphabet letters indicate the significant difference between groups". I could not understand what is the difference and what are the groups mentioned.

**Reply** *technical comment: can easily be clarified. The meaning of alphabets will be explained in detail as 'Statistically significant differences betweeb sample types (runoff, throughfall, forest floor leachates, soil solution) are indicated by different letters in the box plots, significance level of P < 0.05'.*

Fig. 6 – DTN is dissolved total nitrogen?

**Reply** *technical comment: can easily be clarified. Changed to total dissolved nitrogen (TDN) in the manuscript*

---

## Author Comment (AC2)

Lee et al. in this manuscript have tried to understand the effect of storm events on dissolved and particulate carbon and nitrogen in runoff from two different watersheds dominated with different species of vegetation. In general, study of this kind can provide an improved understanding of nutrients and material transfer from different terrestrial set up under monsoon condition. It is understandable that it requires a lot of effort to carry out study of this nature; it would have been better if there were more sampling events. Keeping aside the limitation in numbers of sampling events, the manuscript in present form is very poorly written with lots of mistakes in presentation of results and figures. I believe it is not suitable for publication in Biogeosciences. Below are some of my specific observations:

*Reply We appreciate your valuable comments and suggestions to improve our manuscript. As you indicated, logistical limitations did not allow us to take more storm samples. However, our approach combining elemental concentrations and isotope ratios of both DOM and POM provides rare data sets and insights, which would attract attention from the readership of Biogeosciences, we believe. We will thoroughly revise our manuscript incorporating your critical comments. Specifically, the abstract, introduction, and discussion sections will be rewritten in a more focused way to elaborate on motivations and major findings and their implications.*

\* Abstract does not clearly bring out the findings of the study. It merely describes the variability in results. I think they should modify it to include the processes involved for such observation.

*Reply The abstract will be revised to highlight major findings, including watershed-specific differential storm responses of DOC vs. POC (PON) and DON vs. DIN. We already suggested differences in hydrologic flowpaths as a major mechanism for the differential storm responses observed in the two watersheds. This proposed mechanism will be complemented with more detailed descriptions of the interplay between hydrology and species differences affecting litter and SOM chemistry.*

*The abstract can shortly include processes and explanation for observation as from line 15 on:*

*'During storm events, DOC concentrations in runoff increased with discharge, while DON concentrations were stable. DOC, DON and $NO_3$-N fluxes in runoff increased linearly with discharge pointing to changing flow paths from deeper to upper soil layers at high discharge, whereas nonlinear responses of POC and PON fluxes were observed likely due to the origin of particulate matter from the erosion of mineral soil along the stream benches. The cumulative C and N fluxes in runoff were in the order; DOC > POC probably caused by the relatively moderate precipitation less than 100 mm per event and $NO_3$-N > DON > PON. The cumulative DOC fluxes in runoff during the 2 months study period were much larger at the deciduous watershed (16 kg C ha$^{-1}$) than at the mixed watershed (7 kg C ha$^{-1}$) while the cumulative $NO_3$-N fluxes were higher at the mixed watershed (5.2 kg N ha$^{-1}$) than at the deciduous watershed (2.9 kg N ha$^{-1}$), suggesting a larger N uptake by deciduous trees. Cumulative fluxes of POC and PON were similar at both watersheds. Quality parameters of organic matter in soils and runoff indicate that the contribution of near surface flow to runoff was larger at the deciduous than at the mixed watershed. Our results demonstrate different responses of dissolved C and N in runoff to storm events as a combined effect of tree species composition and watershed-specific flowpaths.*

\* Introduction is poorly written with no focus. It wanders from one topic to other other without gravitating towards the focus of the work. The first line of the introduction itself appears confusing to me.

*Reply The introduction was rather short in the 1. version. We will extend the introduction and cite more previous studies so that the focus of the paper becomes clearer. We will focus on three main topics: heavy rainfall effects, tree species effects and the relation among DOC, DON, DIN, POC, and PON.*

\* One common observation throughout the manuscripts is regarding the references. I

think it should be chronologically arranged.

*Reply We followed manuscript preparation guidelines for authors in Biogeosciences webpage (source: http://www.biogeosciences.net/for_authors/manuscript_preparation.html), which regulate that 'In terms of in-text citations, the order can be based on relevance, as well as chronological or alphabetical listing, depending on the author's preference'. However, we can chronologically arrange the references in text, if editor wants us to cite references chronologically.*

\* Framing of the sentences from previous works is such that they appear as if they are from the present study (line 15-20).

*Reply Line 17 will be changed to 'Only few data were available on the partitioning of DON and PON fluxes in runoff from forested watersheds, like Inamdar et al., (2015)'.*

\* Introduction last sentence: What is measuring campaign?

*Reply technical comment: The term will be deleted. Also, the sentence will be simply written to 'The goal of this study was thus to investigate the influence of tree species and heavy storm events on the fluxes of dissolved and particulate forms of C and N from a mixed coniferous/deciduous and a deciduous forested watershed in South Korea during the 2013 monsoon season'.*

\* How can annual air temperature range from 10 - 15 oC with -6oC in January and 26 oC in August?

*Reply clarified as 'The average annual temperature of the Lake Soyang watershed in western Gangwon-province is $11^oC$ with monthly average temperature ranging from $-5^oC$ in January to $24^oC$ in August during from 1981 to 2010.*

\* Page 3: Line 20: '(deciduous watershed) (Figure 1)' should be replace with (deciduous watershed; Figure 1).

*Reply technical comment: It will be changed to '(deciduous watershed; Figure 1)'.*

Page 4: The text suddenly jumps to Oi and Oe+Oa layers without providing any context to it.

*Reply The definitions of Oi, Oe and Oa will be added: 'The total stock of organic horizons (Oi: slightly decomposed recognizable litter, Oe: moderately decomposed fragmented litter, Oa: highly decomposed humus material) was collected at each plot in a $20 \times 20$ cm frame with 10 replicates'.*

Page 4: What do you mean by partly below detection limit. Please provide the number of samples or occasions when it went below detection limit.

*Reply The detection limits were already stated in the text; page 4 line 24. Concentrations less than detection limit were observed in 5-8% of the measurements in runoff during the July events. This information will be given in the methods section*

\* The authors mention the statistical methods followed for analysis but it hardly comes during discussion.

*Reply Whenever we cite results that are significant, statistical significance will be mentioned throughout the text.*

\* Page 5: Line 25: Elaborate on the meaning of freeze drying the samples for mass spectrometric analysis or rephrase the sentence.

*Reply The sentence at the page 5: line 25 will be changed to 'After filtration (0.45 μm, Whatman), water samples were freeze-dried.*
*Water samples were freeze dried to measure isotope abundance because freeze drying is widely used as pre-treatment of water samples for isotope analysis (Lee et al. 2013, Lamber et al. 2014).*
   *-J.-Y. Lee et al. (2013) Variation in carbon and nitrogen stable isotopes in POM and zooplankton in a deep reservoir and relationship to hydrological characteristics, Journal of Freshwater Ecology, 28(1):47–62.*

- *Lambert T. et al. (2014) DOC sources and DOC transport pathways in a small headwater catchment as revealed by carbon isotope fluctuation during storm events. Biogeosciences, 11:3043-3056.*

\* Should not mineral N include nitrite?
*Reply Nitrite was not measured because it was negligible in soil solutions and runoff from test measurements.*

\* Authors state that 'the soil _13C and soil_15N values significantly increased with soil depth from -29 to -24‰ and from 0 to 8‰ respectively". However, it would be nice to see the vertical profiles of such data. Was the surface _15N always near 0 ‰
*Reply New figure of soil $^{13}$C and $^{15}$N isotope will be added. The added data will be used as evidence of watershed-specific species effect on the quality of soil organic matter.*

[Figure]

\* I could not make the sense out of the sentence 'In the study period, the highest precipitation coincided with the maximum precipitation intensity, the highest precipitation intensity and the maximum discharge at the 10 mixed watershed and at the deciduous watershed on July 14th, 2013"
*Reply technical comment: This sentence will be deleted.*

\* While the difference in DOC concentrations with discharge between deciduous and mixed watershed appears to be convincing (Fig 2a), the POC and PON increase with discharge relies heavily on one data point from high discharge. I do not doubt the increase but I believe that to make unequivocal conclusion more points would have been an asset.
*Reply the discussion will be changed. The "threshold" interpretation will be weakened*

\* Please rephrase the sentence "The fluxes of DOC and NO3-N increased with a much steeper slope at the deciduous and at the mixed watershed, respectively".
*Reply The sentence will be rephrased as 'the DOC fluxes at the deciduous watershed increased with a much steeper slope in response to discharge than at the mixed watershed, while $NO_3$-N fluxes at the mixed watershed steeply increased with increasing discharge than at the deciduous one.*

\* There is problem with the symbols and its representation in Figure 3. I think authors should be careful with these kinds of mistakes before submitting their manuscript for review. It is very tiring to review a manuscript with these kinds of mistakes.
*Reply sorry for the confusion: the mistakes will be corrected in the new figure.*

\* Result section discussing Chemical properties of DOM and POM in runoff should be modified with proper emphasis on isotopic data. At present the isotopic data has just been mentioned as passing comment.

*Reply* *We will provide more detailed descriptions on the isotopic signatures in the discussion section on Chemical properties of DOM and POM in runoff: `Also the $^{13}C$ data in runoff, being more negative at the deciduous watershed points to a larger proportion of forest floor leachates than at the coniferous watershed'.*

\* In the discussion, authors have admitted that the numbers of events are rather low in the study and observations made by them have already been observed before by Dhillon and Inamdar (2013). I am wondering what novel finding they are discussing which warrants publication in a journal like Biogeosciences.

*Reply* *Most of previous studies focused on the fluxes of organic matter at one watershed for one year or more. In our case, the novelty lies in comparing differential storm responses of DOC/DON and POC/PON with a particular reference to watershed properties and storm response patterns.*

\* Discussion section needs to be re-written with proper emphasis on the major findings C3 from this work. The mechanisms and processes behind the differences in observation need to be discussed properly. The effect of altitude, nature of littler and specific nature of the two watersheds needs to be take in account.

*Reply* *The discussion will be rewritten to provide more in-depth discussions of the three main points of the manuscript (species as a key watershed characteristic determining storm responses, differential responses of DOM vs. POM, and differential responses of DON vs DIN) and also mechanistic interpretation of the findings. We will also address other watershed characteristics as you suggested (altitude, nature of litters and specific nature of the two watersheds) for example, 'The deciduous watershed is located at higher altitude suggesting more shallow soils than at the mixed watershed. This may explain the larger near surface flow paths at the deciduous watershed. Moreover, faster decomposition of the deciduous litter leaches relatively more DOM and both factors resulted in higher DOC export fluxes at the deciduous watershed than at the mixed watershed'.*

\*Fig 2: What are FPOC/FPON?
*Reply* *technical comment: can easily be clarified. Will be changed to POC/PON.*

Fig 5: A succinct Fig 5 will be better.

*Reply* *The design of figure 5 and 6 can be changed. Please check below figures how it will be changed 1) the compartments at the y axis are rearranged downwards. 2) the x axis description is only once to the two graphs 3) PLF/FLF in figure 5 is removed.*

[Figure]

Fig 5 new

[Figure]

Fig. 6 new

---

## Author Response (AR1)

**Comments from referees**

*In blue font color the reply of the authors to the referee comments with line numbers referring to the manuscript version with the yellow-marked changes*

General comments

The manuscript presented very useful information on forms and flows of carbon and nitrogen in an upland fluvial system. The planning and coverage of sampling and parameters are good. The results will improve our understanding of

10   material flows in terrestrial fluvial systems based on nature of tree types and water flows. The results deserve to be published but not in the present form. The basic problem is with the presentation and the way they dealt with scientific and technical issues. I recommend encouraging the authors for submitting a revised version after they work on some of the issues mentioned and improve the presentation for clarity.

Specific Comments

Title: manuscript is not really studying the 'response of carbon and nitrogen components IN RUNOFF to storm events' but addresses the influence of storms on carbon and nitrogen components in runoff. The appropriate title for the

20   manuscript seems to be (from lines 32-33 on page 2): Influence of tree species and episodic discharges on the fluxes of dissolved and particulate carbon and nitrogen from two watersheds OR Changes in fluxes of dissolved and particulate carbon and nitrogen from two watersheds of different tree types during heavy discharge periods.

*Reply: title has been changed accordingly to 'Changes in fluxes of dissolved and particulate carbon and nitrogen*

25   *from two watersheds of different tree species during intense storm events'.*

Page 3: Lines 4-5 – The sentence 'the annual air temperature ranges from 10-15oC with -6oC in January and 26oC in August' does not make sense to me.

*Reply: it has been clarified as 'The average annual temperature of the Lake Soyang watershed in western*

30   *Gangwon-province is $11^oC$ with monthly average temperature ranging from $-5^oC$ in January to $24^oC$ in August'.*

*See line 11-13 on page 3.*

Page 3: Line 7 –Is the 47% broadleaved forest 'the deciduous'? Here the comparison is between deciduous and mixed types and so appropriate type to be named than the

35   description (broadleaved).

*Reply: it has been clarified as 'Korean mountainous forests are mostly composed of deciduous forests*

*representing 47% of the total forested area (38% coniferous forest, 12% mixed coniferous and deciduous forest)'.*

*See line 15-16 on page 3.*

40   Page 3: Lines 12-14 – Are the slopes at two sampling points MC and MD oriented in different directions in the mixed watershed? If they are oriented in one direction then fluxes from the upper can definitely influence the other during floods. This question also pertains to slope comparisons between two watershed sampling points. Figure 1 and info on page 3 shows that deciduous sampling point is at a higher altitude than that

45   in the mixed watershed. What if both the watersheds (and hence the sampling points) slope in the same direction? If yes, the flow from deciduous sampling point would

influence the composition at MC and MD!! This is quite possible as the two watersheds are nearby. The authors should clarify on this issue of slopes and possible interference between sampling points.

*Reply: we have described the effect of slope direction in the methods: new text: 'The slope direction of the coniferous part at the mixed watershed is towards the MD plot. Lateral flow from the coniferous part to the MD plot can only influence deeper soil solution characteristics as near surface flow was never observed. Our data (see results) indicate significant quality differences of soil solutions between the MD and MC plots which suggest only a minor influence on soil solution chemistry at the MD plot from lateral flows. Furthermore, the quality parameters of soil solutions at the MD plot were similar to those of the DD plot, the latter being not influenced by lateral flows from coniferous sites. Thus, it is unlikely that the MC plot did affect the MD plot'. See line 27-32 on page 3.*

Page 4: Line 5 – 'were collected after each storm event': maximal flows/fluxes must have occurred during the peak flow. When the maximal speeds subsided the original peak signals (of concentrations/fluxes) of the flood may have been lost!! This can be

exemplified using the data in Table 2 for deciduous station. On July 8 (say first flood studied) DOC, DON, POC, PON values are higher than the following flood event on 14 July. Obviously, the first flood water carried more C & N than the second one since the first/fresh rain/flood can dissolve/scoop more of materials accumulated during the preceding dry or intervening periods in the soils. This was also noticed by the authors on Page 6 Line 15.

*Reply: we have clarified this as 'During storm events in July 2013, throughfall, forest floor leachate and soil solution was collected after each storm event so that these samples represent cumulative water samples during the entire storm event'. See line 17-19 on page 4.*

Page 4: Line 5 – 'runoff samples were collected every 1 or 2 h in the weir' – were these also collected after the storm events (coinciding with throughfall, forest floor leachate and soil solution sampling) or during the event or both? This information is crucial for making the right comparisons and assessing changes.

*Reply: we have clarified this as 'In case of runoff, samples were taken in July 2013 at the weir using automatic collectors (6712 Portable Sampler, Teledyne Isco Inc., Lincoln, NE, USA) before, during, and after each rain event at intervals of 1 or 2 h'. See line 19-20 on page 4.*

Page 6: line 3 – In the absence of clear definitions of Oi, Oe and Oa it is hard to understand the significance of percentages of these fractions, as also in relevant Figures.

*Reply: the definition and thickness of horizons have been given: new text: '(Oi: slightly decomposed recognizable litter, Oe: moderately decomposed fragmented litter, Oa: highly decomposed humus material)' and 'The average thickness of Oi and Oe+Oa was 1.2 and 1.5 cm at the MC plot, 2.5 and 3 cm at the MD plot, and 2.3 and 2 cm at the DD plot, respectively', See line 28-29 and line 29-31 on page 4.*

Page 6: Lines 17-21 – I am not convinced of the 'threshold value' since there are not enough data points to show a consistent increase in concentrations. Relatively higher concentrations in POC and PON are found (Fig 2d,e) RANDOMLY during discharges from _1 to 9 mm/h.

*Reply: the reviewer is right, text has been changed accordingly to 'At discharges from ~1 to 9 mm $h^{-1}$, higher*

*concentrations of POC and PON in runoff were found (Figure 3d,e). For example, the POC concentration in runoff from the mixed watershed was as high as 10.7 mg C L$^{-1}$ at the largest discharge of 9 mm h$^{-1}$. At the deciduous watershed, the POC concentration in runoff reached a maximum of 8.6 mg C L$^{-1}$ already at 3 mm h$^{-1}$ discharge during the first storm event (Figure 3d, Table 2)'. See line 7-10 on page 7.*

Page 6: Lines 22-23 – DOC rise with increasing flood and fall with decreasing flood is convincing and is, indeed a good observation.
***Reply:*** *thank you for your comment; no reply needed.*

10 Page 8: in the entire page of this discussion, the authors did not seem to have paid much attention to (a) nature of litter, (b) altitude and (c) substratum of the two watersheds. I understood that the deciduous watershed was at higher altitude with hard rock below 40 cm whereas that of mixed was at lower height laden with soils upto or below 50 cm. Presumably the hard rock might have occurred deeper in the mixed watershed.
15 The nature off litter (seasonally fallen parts of the trees) would be relatively freshly fallen in the deciduous watershed that could easily be broken/decomposed by physical/microbial activities that could leach more DOC or dissolvable OM. This fresh DOM can be easily be flushed by flowing water. The hard substratum and high altitude facilitate flow of water at higher speeds (as it cannot seep deep) in the deciduous wa-tershed than in the mixed
20 watershed. The rapidly draining flood facilitates easy mixing of forest floor and soil solutions with surface runoff.
Although logical in their statement of "i) In the deciduous litter layer the leaves are overlapping and are partly impermeable which may cause more surface near flow (lines 23-25)" this does support their observation that 'a larger proportion of the DOC in runoff
25 results from forest floor leachates at the deciduous (lines 17-18)'. If the top layer is impermeable how would one explain high DOC in runoff to have come from mixing with forest floor leachates? It is also possible for high DOC formation at the surface itself as the fresher litter is weathered or decomposed on the floor of the deciduous watershed.
30 ***Reply:*** *we agree with the last statement and our argumentation in the discussion has been modified for more clarity. Your comments have been incorporated in more detailed discussion. New text: 'As the deciduous watershed is located at a higher altitude the soils might be more shallow than at the mixed watershed which will add to the larger near surface flow paths'. and 'Faster decomposition of the deciduous litter leaches relatively more DOM and both factors result in higher DOC export fluxes at the deciduous than at the mixed watershed.*
35 *Based on our data set of this study, one cannot quantify the relative importance of these factors for the differences between the watersheds'. See line 20-22 and line 23-26 on page 9.*

Page 9: Lines 2-8 – higher DOC/DON ratio at deciduous basin is possible when organic matter with less or no nitrogen but rich in carbon is weathered and is leached.
40 Nitrogen compounds are perhaps enriched in litter or particulates. However, Fig 2 depicts lower PON than DON, in general in both watersheds, implying that nitrogen might be remaining with the deposited litter in the watersheds!
***Reply:*** *we explained the larger DOC/DON ratios at the deciduous watershed in lines 14-16 on page 8. The mobilization of particulate organic matter is attributed to the erosion or river benches and no conclusion on N*
45 *retention in litter is possible. See line 17-18 on page 10.*

Page 9: Statements in lines 11-13 ('Substantial fluxes of NO3-N and the dominance of NO3-N over DON in runoff are likely due to a certain degree of N-saturation (N supply > N demand) of these forested watersheds (Aber et al., 1998; Compton et al., 2003)') and lines 20-21 ('Overall, it seems that a larger N uptake by the deciduous trees at the deciduous watershed could explain the differences in NO3-N fluxes') are arbitrary and not supported by any data.

*Reply: true, the conclusion is speculative and that is why it is formulated as a suggestion. However, other reasons for the higher $NO_3$ fluxes at the coniferous site are not likely.*

Technical Comments Page 3: Lines 11 and 21 – Are the latitude and longitude positions accurate to the decimals mentioned?

*Reply: the positions of watershed at the weir have been corrected as 38°12´N, 128°11´E for the mixed watershed in Seohwa and 38°15´N, 128°7´E for the deciduous watershed in Haean. See line 19 on page 3 and line 1-2 on page 4.*

Page 3: Line 30: Does 'throughfall' refer to precipitation or rainwater?

*Reply: the sentence has been written as 'Throughfall collectors (n=5) under the canopy were equipped with filters to prevent large particles from entering'. See line 12 on page 4.*

Page 4: Line 13 – define Oi, Oe and Oa.

*Reply: as mentioned earlier, the definitions of Oi, Oe and Oa have been added in line 28-29 on page 4.*

Page 4: Lines 26-27 – 'The storm events during monsoon season were identified from the start to the end of precipitation with more than a day interval between each storm event'. The storm events should be identified based on meteorological observations of wind and rainfall. However, one should keep in mind that the present study is made in summer monsoon. During monsoon season the rainfall may not be continuous on all days but with intermittent gaps (breaks) or spells of rain. I guess the authors are referring to these spells, or at the most the episodic rainfall events (which are normal during summer monsoon) of variable duration as 'storms'. This requires authors clarification for what they meant by 'storm'. This point, however, was rightly stated by the authors on page 8 line 4 – 'four heavy rainfall events of the monsoon season at both watersheds' but not elsewhere in the manuscript.

*Reply: as you suggested, the rainfall characteristic during monsoon has been referred as 'During the monsoon season the rainfall was not continuous on all days but with intermittent gaps. The most lasting rainfall events were identified as storm events with more than a day interval between each storm event'. The term 'rainfall' has been used to explain 'storms' in the method part of this manuscript. See line 13-15 on page 5.*

Page 5: Lines 4 and 8 – (i) unmatched DOC and POC cutoff limits! (ii) 0.7 micron cutoff limit for POC is quite on higher side since most of the fine sized particulate materials are lost through the filter paper.

*Reply: we do not agree that the fraction from 0.45 to 0.7 micron represents most of the fine sized material. DOM is commonly defined as organic matter in water samples smaller than 0.45µm (Thurman, 1985). Previous studies have often used a 0.7 µm pore size of glass filter for POM fraction for technical aspects in the analysis (Bauer and Bianchi 2011, Mostofa et al. 2013). Consequently, DOC and POC cutoff limits are unmatched as you pointed. However, prior tests (Doyle 2013) showed that materials between 0.45 and 0.7 µm comprised a minor fraction in*

*total organic matter.*

*Therefore, we have made a comment on that in the methods section: new text: 'DOC and POC cutoff limits as 0.45 and 0.7 μm were unmatched in this study because of practical reasons and the unmatched fraction is considered negligible'. See line 2 on page 6.*

-Thurman, E. M. (1985). Organic geochemistry of natural waters. Nordrecht, The Netherlands: Martinus Nijhoff/Junk Publisher.
-Bauer, J.E., and Bianchi, T.S. (2011). 5.02—dissolved organic carbon cycling and transformation. Treatise on estuarine and coastal science, 5, 7-67.
-Mostofa, K.M., Liu, C.Q., Minakata, D., Wu, F., Vione, D., Mottaleb, M.A., ... and Sakugawa, H. (2013). Photoinduced and Microbial Degradation of Dissolved Organic Matter in Natural Waters. In Photobiogeochemistry of Organic Matter. Springer Berlin Heidelberg, 273-364
-Doyle, C. B. (2013). Contribution of bacterial cells to the fluorescence spectra of natural organic matter in freshwaters, University of North Carolina, master thesis.

Page 5: Line 6 – was nitrite in water analysed? It should be included in mineral-N.
*Reply: nitrite was not measured because it was negligible in soil solutions and runoff from test measurement. We have made a comment on that in the methods section: new text: 'Nitrite was not measured because concentrations were negligible in soil solutions and runoff'. See line 24-25 on page 5.*

Fig 2 – what are FPOC/FPON in Fig. 2f?
*Reply: technical comment: it has been changed to POC/PON in figure 3 on page 19.*

Fig. 3. Upper panel in the left column – DON and PON should be corrected to DOC and POC.
*Reply: technical comment: it has been corrected to DOC and POC in figure 4 on page 20.*

Figures 5 & 6: Alphabets (a, b: : :.) need to be explained in more detail. For instance what does it mean by ab or abcd. In the captions it is mentioned "Different alphabet letters indicate the significant difference between groups". I could not understand what is the difference and what are the groups mentioned.
*Reply: technical comment: the meaning of alphabets has been explained in detail as 'Statistically significant differences between sample types (throughfall, forest floor leachates, soil solution, and runoff) are indicated by different letters in the box plots, significance level of $p<0.05$'. See in caption of figure 6 and 7 on page 23 and 24, respectively.*

Fig. 6 – DTN is dissolved total nitrogen?
*Reply: technical comment: it has been changed to total dissolved nitrogen (TDN) in the manuscript.*

**Anonymous Referee #1**

Lee et al. in this manuscript have tried to understand the effect of storm events on
dissolved and particulate carbon and nitrogen in runoff from two different watersheds
dominated with different species of vegetation. In general, study of this kind can provide
an improved understanding of nutrients and material transfer from different terrestrial
set up under monsoon condition. It is understandable that it requires a lot of effort to
carry out study of this nature; it would have been better if there were more sampling
events. Keeping aside the limitation in numbers of sampling events, the manuscript in
present form is very poorly written with lots of mistakes in presentation of results and
figures. I believe it is not suitable for publication in Biogeosciences. Below are some of my specific observations:

* Abstract does not clearly bring out the findings of the study. It merely describes the
variability in results. I think they should modify it to include the processes involved for
such observation.

*Reply: the abstract has been revised to highlight major findings, including watershed-specific differential storm responses of DOC vs. POC (PON) and DON vs. DIN. We already suggested differences in hydrologic flow paths as a major mechanism for the differential storm responses observed in the two watersheds. This proposed mechanism has been complemented with more detailed descriptions of the interplay between hydrology and species differences affecting litter and SOM chemistry. See the changes in line 17-27 in the abstract.*

* Introduction is poorly written with no focus. It wanders from one topic to other other
without gravitating towards the focus of the work. The first line of the introduction itself
appears confusing to me.

*Reply: the introduction has been extended with connections and references so that the focus of the paper became clearer. The first line of the introduction has been deleted. We have focused on three main topics: heavy rainfall effects, tree species effects and the relation among DOC, DON, DIN, POC, and PON.*

* One common observation throughout the manuscripts is regarding the references. I think it should be
chronologically arranged.

*Reply: we followed manuscript preparation guidelines for authors in Biogeosciences webpage (source: http://www.biogeosciences.net/for_authors/manuscript_preparation.html), which regulate that 'In terms of in-text citations, the order can be based on relevance, as well as chronological or alphabetical listing, depending on the author's preference'.*

* Framing of the sentences from previous works is such that they appear as if they are
from the present study (line 15-20).

*Reply: this sentence has been changed to 'Only few data are available on the partitioning of DON and PON fluxes in runoff from forested watersheds, like Inamdar et al. (2015)'. See line 20-22 on page 2.*

* Introduction last sentence: What is measuring campaign?

*Reply: technical comment: the term has been deleted. Also, the sentence has been simply written to 'The goal of this study was thus to investigate the influence of tree species and heavy storm events on the fluxes of dissolved and particulate forms of C and N from a mixed coniferous/deciduous and a deciduous forested watershed in South Korea during the 2013 monsoon season'. See line 4-7 on page 3.*

\* How can annual air temperature range from 10 - 15 oC with -6oC in January and 26
oC in August?
*Reply: it has been clarified as 'The average annual temperature of the Lake Soyang watershed in western*
*Gangwon-province is 11°C with monthly average temperature ranging from -5°C in January to 24°C in August'.*
*See line 11-13 on page 3.*

\* Page 3: Line 20: '(deciduous watershed) (Figure 1)' should be replace with (deciduous
watershed; Figure 1).
*Reply: technical comment: the format has been changed to '(mixed watershed; Figure 1) or '(deciduous*
*watershed; Figure 1)'. See line 18 and 1 on page 3 and 4, respectively.*

Page 4: The text suddenly jumps to Oi and Oe+Oa layers without providing any context
to it.
*Reply: the definition of horizons has been given: new text: '(Oi: slightly decomposed recognizable litter, Oe:*
*moderately decomposed fragmented litter, Oa: highly decomposed humus material)'. See line 28-29 on page 4.*

Page 4: What do you mean by partly below detection limit. Please provide the number
of samples or occasions when it went below detection limit.
*Reply: the detection limits were already stated in the text; line 11 on page 5. The new text in the methods section*
*has been given as 'Concentrations less than detection limit were observed in 5-8% of the measurements in runoff*
*during the July events'. See line 9 on page 5.*

\* The authors mention the statistical methods followed for analysis but it hardly comes
during discussion.
*Reply: the statistical significance was already mentioned for figure 6 and 7. See the caption of figure 6 and 7 on*
*page 23 and 24.*

\* Page 5: Line 25: Elaborate on the meaning of freeze drying the samples for mass
spectrometric analysis or rephrase the sentence.
*Reply: the sentence has been changed to 'After filtration (0.45 μm, Whatman), water samples were freeze-dried*
*to measure $^{13}C$ and $^{15}N$ isotope abundances of DOC and TDN…'. See line 14-15 on page 6.*
*Water samples were freeze dried to measure isotope abundance because freeze drying is widely used as pre-*
*treatment of water samples for isotope analysis (Lee et al. 2013, Lamber et al. 2014).*
*-J.-Y. Lee et al. (2013) Variation in carbon and nitrogen stable isotopes in POM and zooplankton in a deep*
*reservoir and relationship to hydrological characteristics, Journal of Freshwater Ecology, 28(1):47–*
*62.*
*- Lambert T. et al. (2014) DOC sources and DOC transport pathways in a small headwater catchment as*
*revealed by carbon isotope fluctuation during storm events. Biogeosciences, 11:3043-3056.*

\* Should not mineral N include nitrite?
*Reply: nitrite was not measured because it was negligible in soil solutions and runoff from test measurement. We*
*have made a comment on that in the methods section: new text: 'Nitrite was not measured because*
*concentrations were negligible in soil solutions and runoff'. See line 24-25 on page 5.*

\* Authors state that 'the soil _13C and soil_15N values significantly increased with soil depth from -29 to -24‰

and from 0 to 8‰ respectively". However, it would be nice to see the vertical profiles of such data. Was the surface _15N always near 0 ‰

*Reply: New figure of soil profiles of $^{13}C$ and $^{15}N$ isotope abundance has been added. See new figure 2 on page 18*

5   \* I could not make the sense out of the sentence 'In the study period, the highest precipitation coincided with the maximum precipitation intensity, the highest precipitation
intensity and the maximum discharge at the 10 mixed watershed and at the deciduous
watershed on July 14th, 2013"
*Reply: technical comment: this sentence has been deleted.*

\* While the difference in DOC concentrations with discharge between deciduous and mixed watershed appears to be convincing (Fig 2a), the POC and PON increase with discharge relies heavily on one data point from high discharge. I do not doubt the increase but I believe that to make unequivocal conclusion more points would have
15  been an asset.
*Reply: the discussion was changed. The "threshold" interpretation has been weakened. See the discussion section 4.3 on page 10.*

\* Please rephrase the sentence "The fluxes of DOC and NO3-N increased with a much
20  steeper slope at the deciduous and at the mixed watershed, respectively".
*Reply: the sentence was rephrased as 'The DOC fluxes at the deciduous watershed increased with a much steeper slope in response to discharge than at the mixed, while the $NO_3$-N fluxes at the mixed watershed more steeply increased with increasing discharge than at the deciduous'. See line 21-23 on page 7.*

25  \* There is problem with the symbols and its representation in Figure 3. I think authors should be careful with these kinds of mistakes before submitting their manuscript for review. It is very tiring to review a manuscript with these kinds of mistakes.
*Reply: sorry for the confusion: the mistakes have been corrected in the new figure.*

30  \* Result section discussing Chemical properties of DOM and POM in runoff should be modified with proper emphasis on isotopic data. At present the isotopic data has just been mentioned as passing comment.
*Reply: we have descried the isotopic signatures in the result section: new text: 'Also, the $^{13}C$ data in runoff, being*

*more negative at the deciduous watershed, point to a larger proportion of forest floor leachates in runoff than at*

35  *the coniferous watershed'. See line 23-13 on page 8.*

\* In the discussion, authors have admitted that the numbers of events are rather low in the study and observations made by them have already been observed before by Dhillon and Inamdar (2013). I am wondering what novel finding they are discussing which warrants publication in a journal like Biogeosciences.
40  *Reply: Most of previous studies focused on the fluxes of organic matter at one watershed for one year or more. In our case, the novelty lies in comparing differential storm responses of DOC/DON and POC/PON with a particular reference to watershed properties and storm response patterns.*

\* Discussion section needs to be re-written with proper emphasis on the major findings

C3 from this work. The mechanisms and processes behind the differences in observation need to be discussed properly. The effect of altitude, nature of littler and specific nature of the two watersheds needs to be take in account.

*Reply: in the discussion, we have addressed other watershed characteristics as you suggested (altitude, nature of litters and specific nature of the two watersheds) for example, v) As the deciduous watershed is located at a higher altitude the soils might be more shallow than at the mixed watershed which will add to the larger near surface flow paths. vi) Faster decomposition of the deciduous litter leaches relatively more DOM and both factors result in higher DOC export fluxes at the deciduous than at the mixed watershed. Based on our data set of this study, one cannot quantify the relative importance of these factors for the differences between the watersheds'. See line 20-26 on page 9.*

*Fig 2: What are FPOC/FPON?
*Reply: technical comment: it has been changed to POC/PON in figure 3 on page 19.*

Fig 5: A succinct Fig 5 will be better.

*Reply: The design of the figure has been changed, especially 1) the compartments at the y axis are rearranged downwards. 2) the x axis description is only once to the two graphs 3) PLF/FLF in figure 5 is removed. See new figure 6 and*

[revised manuscript text omitted]

---

## Editor Decision (ED1)

**Reference: Manuscript # bg-2016-92; corrected version of abstract**

**Abstract.**  This study  reports on  large variability associated with intense storm events on the runoff fluxes and  composition of dissolved (< 0.45 μm) and particulate (0.7 μm to 1 mm) organic carbon and nitrogen (DOC, DON, POC, PON) in a mixed coniferous/deciduous (mixed watershed) and a deciduous forested watershed (deciduous watershed) in South Korea. During storm events, DOC concentration in runoff increased with water discharge  in contrast to more or less uniform concentration observed for DON. . DOC, DON and NO$_3$-N runoff fluxes  increased linearly with discharge pointing to changing flow paths from deeper to upper soil layers at high discharge conditions; whereas nonlinear response was observed for  POC and PON fluxes  suggesting  origin of particulate matter from the erosion of  soil along the stream branches . **[Check: Is it branches or benches?]** The cumulative C and N fluxes   followed the order as: DOC > POC and NO3$^-$-N > DON > PON. The cumulative DOC fluxes in runoff during the  study period of two months were much  higher at the deciduous watershed (16 kg C ha$^{-1}$) than at the mixed watershed (7 kg C ha$^{-1}$), while the cumulative NO$_3$-N fluxes were higher at the mixed watershed (5.2 kg N ha$^{-1}$) than at the deciduous watershed (2.9 kg N ha$^{-1}$). The latter observation suggests  significantly large N uptake by deciduous trees. Cumulative fluxes of POC and PON were similar at both watersheds.  Composition of organic matter in soils and runoff indicate that the contribution of near surface flow to runoff was  higher at the deciduous than at the mixed watershed. Our results demonstrate different response  on particulate and dissolved C and N  fluxes  during storm events  arising from combination of factors  related to composition of tree species  and watershed specific flow paths.

---

## Author Response (AR2)

Response to the comments of the editor:

There is still lack of clarity in the revised abstract of the manuscript submitted by Authors. As pointed out under specific comments, some of the sentences and words are rather confusing. A corrected version of the abstract is attached here. Authors may like to consider the changes as appropriate and agreeable to improve

5    the clarity.

Specific comments:

1) First two sentences of the abstract are redundant and should be deleted.

Reply: we deleted the second sentence. The first is needed as a general rationale of the study.

2) Abstract, 3rd line: What authors mean by the phrase "on the quality of dissolved -------. Are they referring to composition of dissolved and particulate OC and ON?

Reply: changed to "composition of"

15    3) Abstract, 6th line: "------ concentrations were stable". This is not a right phrase to use. See the corrected abstract for better clarity.

Reply: changed to "concentrations remained almost constant"

4) Abstract, 9th line: Is it stream benches or "branches". Please check!

20    Reply: benches was o.k.

5) Abstract, 10th and 13th line: What is cumulative fluxes? Are these integrated over the study period of two months? Pl clarify!

Reply: cumulative was changed to integrated "The integrated C and N fluxes in runoff over the 2 months

25    study period…"

6) Abstract, 14th line: What authors mean by the phrase "Quality parameters of organic matter in soils and runoff --------". Are they referring to composition of organic matter? See corrected abstract for clarity.

Reply: changed to "the composition of organic matter…"

30

7) Manuscript title should be changed to "Variability in runoff fluxes of dissolved------------------ during intense storm events".

Reply: title was changed accordingly

[revised manuscript text omitted]